# Elevated FBXO45 promotes liver tumorigenesis through enhancing IGF2BP1 ubiquitination and subsequent PLK1 upregulation

**Xiao-Tong Lin, Hong-Qiang Yu, Lei Fang, Ye Tan, Ze-Yu Liu, Di Wu, Jie Zhang, Hao-Jun Xiong, Chuan-Ming Xie***

Key Laboratory of Hepatobiliary and Pancreatic Surgery, Institute of Hepatobiliary Surgery, Southwest Hospital, Third Military Medical University (Army Medical University), Chongqing, China

**Abstract** Dysregulation of tumor-relevant proteins may contribute to human hepatocellular carcinoma (HCC) tumorigenesis. FBXO45 is an E3 ubiquitin ligase that is frequently elevated expression in human HCC. However, it remains unknown whether FBXO45 is associated with hepatocarcinogenesis and how to treat HCC patients with high FBXO45 expression. Here, IHC and qPCR analysis revealed that FBXO45 protein and mRNA were highly expressed in 54.3% (57 of 105) and 52.2% (132 of 253) of the HCC tissue samples, respectively. Highly expressed FBXO45 promoted liver tumorigenesis in transgenic mice. Mechanistically, FBXO45 promoted IGF2BP1 ubiquitination at the Lys190 and Lys450 sites and subsequent activation, leading to the upregulation of PLK1 expression and the induction of cell proliferation and liver tumorigenesis in vitro and in vivo. PLK1 inhibition or IGF2BP1 knockdown significantly blocked FBXO45-driven liver tumorigenesis in FBXO45 transgenic mice, primary cells, and HCCs. Furthermore, IHC analysis on HCC tissue samples revealed a positive association between the hyperexpression of FBXO45 and PLK1/IGF2BP1, and both had positive relationship with poor survival in HCC patients. Thus, FBXO45 plays an important role in promoting liver tumorigenesis through IGF2BP1 ubiquitination and activation, and subsequent PLK1 upregulation, suggesting a new strategy for treating HCC by targeting FBXO45/IGF2BP1/PLK1 axis.

**\*For correspondence:**
chuanming506@126.com

## Introduction

Hepatocellular carcinoma (HCC) is a highly aggressive primary liver malignancy. HCC treatment strategy is based on tumor staging. Staging system such as TNM (tumor-node-metastasis) staging has been proposed for HCC, where T represents the characteristics of the primary tumor, N describes the presence or absence of spread to certain lymph nodes, and M details whether the tumor has distant metastasis in the body. According to the eighth American Joint Committee on Cancer (AJCC) TNM Staging for HCC, T1 (solitary tumor ≤2 cm; solitary tumor >2 cm without vascular invasion), T2 (solitary tumor >2 cm with vascular invasion; or multiple tumors, none >5 cm), T3 (multiple tumors, at least one of which is >5 cm), and T4 (single tumor or multiple tumors of any size involving a major branch of the portal vein or hepatic vein, or tumor(s) with direct invasion of adjacent organs other than the gallbladder or with perforation of visceral peritoneum) were differentiated. With these descriptors, stages I, II, III, and IV were defined as T1 N0 M0, T2 N0 M0, T3 N0 M0, and T4 N0 M0 or any T N1 M0 or any T Any N M1, respectively.

Although treatment strategies against HCC have developed in recent years, such as the application of molecule-targeted drug, yet it is still one of the leading causes of cancer-related mortality

worldwide. Currently, around 60% of HCC patients are diagnosed at advanced stages of carcinogenesis, consequently not able to receive potentially curative treatments (*Altekruse et al., 2009*; *Cucchetti et al., 2013*; *Kanwal and Singal, 2019*; *Llovet et al., 2019*). The pathogenesis of HCC is poorly understood. Therefore, more efforts should be devoted to investigating the molecular mechanisms of HCC tumorigenesis and development. Mounting evidence shows that HCC is a stepwise process, as HCC is a heterogeneous disease driven by the serial accumulation of mutations in tumor suppressor genes (such as *TP53, PTEN,* and *RB*) and proto-oncogenes (such as *MYC, MET, BRAF,* and *RAS*)(*Zhang et al., 2020*). Thus, it is especially important to expound the possible molecular mechanisms underlying the development of HCC.

Ubiquitination is a posttranslational modification that can affect protein localization and regulate cell cycle progression, apoptosis, and transcriptional activity (*Teixeira and Reed, 2013*; *DiAntonio et al., 2001*; *Meyer-Schwesinger, 2019*). The ubiquitin-proteasome system (UPS) is initiated by the conjugation of ubiquitin (a 76-amino acid polypeptide) to a substrate protein, which involves enzymes in three distinct classes of enzymes, a ubiquitin-activating enzyme (E1), ubiquitin-conjugating enzyme (E2), and ubiquitin ligase (E3)(*Micel et al., 2013*). E3 ubiquitin-protein ligases are critical in the UPS for the selectivity needed to target specific proteins for degradation (*Deshaies and Joazeiro, 2009*; *Nakayama and Nakayama, 2006*). Increased evidence has indicated that HCC is associated with abnormal regulation of E3 ubiquitin ligases (*Shen et al., 2018*; *Li et al., 2014*). F-box protein is the substrate recognizing subunit of SCF (SKP1, CUL1, and F-box protein) E3 ubiquitin ligase complexes (*Chen et al., 2014*) and plays important roles in cancer development and progression via regulation of the expression and activity of oncogenes and tumor suppressor genes (*Randle and Laman, 2016*).

F-box/SPRY domain-containing protein 1 (FBXO45) is an atypical E3 ubiquitin ligase (*Saiga et al., 2009*). FBXO45 promotes ubiquitin-dependent proteolysis of some key molecules governing cell survival or DNA damage, in which include FBXW7 (*Richter et al., 2020*), PAR4 (prostate apoptosis response protein 4) (*Chen et al., 2014*), p73 (*Peschiaroli et al., 2009*), and ZEB1 (*Wu et al., 2019*) and it can also block the proteolysis of some proteins, including N-cadherin (*Chung et al., 2014*). Previous studies mainly investigated the role of FBXO45 in cell fate at the cell line level or clinical outcomes in human cancer tissues. Recently, IHC staining showed that nearly 54.2% of HCC patients had high FBXO45 expression in HCC tissues compared with adjacent normal tissues. It is not yet known whether FBXO45 is related to HCC tumorigenesis in vivo or if it has clinical significance in HCC.

Insulin-like growth factor 2 mRNA-binding protein 1 (IGF2BP1) is an oncogenic protein expressed in various cancers, including HCC. For example, IGF2BP1 affects the proliferation and tumorigenic potential of leukemia cells through the critical regulators of self-renewal HOXB4 and MYB and the aldehyde dehydrogenase ALDH1A1 (*Elcheva et al., 2019*). IGF2BP1 increases melanoma metastasis via extracellular vesicle-mediated promotion (*Ghoshal et al., 2019*). IGF2BP1 enhances the expression of various serum response factor (SRF) target genes, including *PDLIM7, FOXK1, MKI67,* and *MYC*, and promotes tumor cell growth and invasion in HCC and other cancers (*Gutschner et al., 2014*; *Müller et al., 2018*). However, a few recent studies found that suppressive roles for IGF2BP1 in tumor growth and metastasis in breast cancer and colon cancer (*Wang et al., 2016*; *Hamilton et al., 2015*).

Here, hyperexpression of FBXO45 was correlated with poor prognosis in HCC. Furthermore, FBXO45 significantly promoted HCC tumorigenesis in FBXO45-OE (OE as over-expressed) mice. Mechanistically, FBXO45 specifically bound to IGF2BP1 and promoted its ubiquitination and activation, followed by upregulation of PLK1. IGF2BP1 was identified as a novel substrate of FBXO45. Targeting IGF2BP1-PLK1 signaling significantly attenuated HCC tumorigenesis triggered by FBXO45. These data uncovered an important role of FBXO45 in regulating HCC tumorigenesis through activation of IGF2BP1-mediated protumorigenic signaling PLK1, suggesting a new strategy for HCC therapy by targeting IGF2BP1-PLK1 axis in HCC patients exhibiting high FBXO45 expression.

## Results

### FBXO45 is associated with poor survival in HCC patients

To determine the role of FBXO45 in human HCC, the protein expression of FBXO45 in a cohort of 105 paired human HCC and adjacent noncancerous liver tissue specimens were examined. IHC staining results showed that FBXO45 expression was significantly increased in 54.3% (57/105) of the HCC samples compared with the adjacent noncancerous tissue samples (*Figure 1A*). This high expression

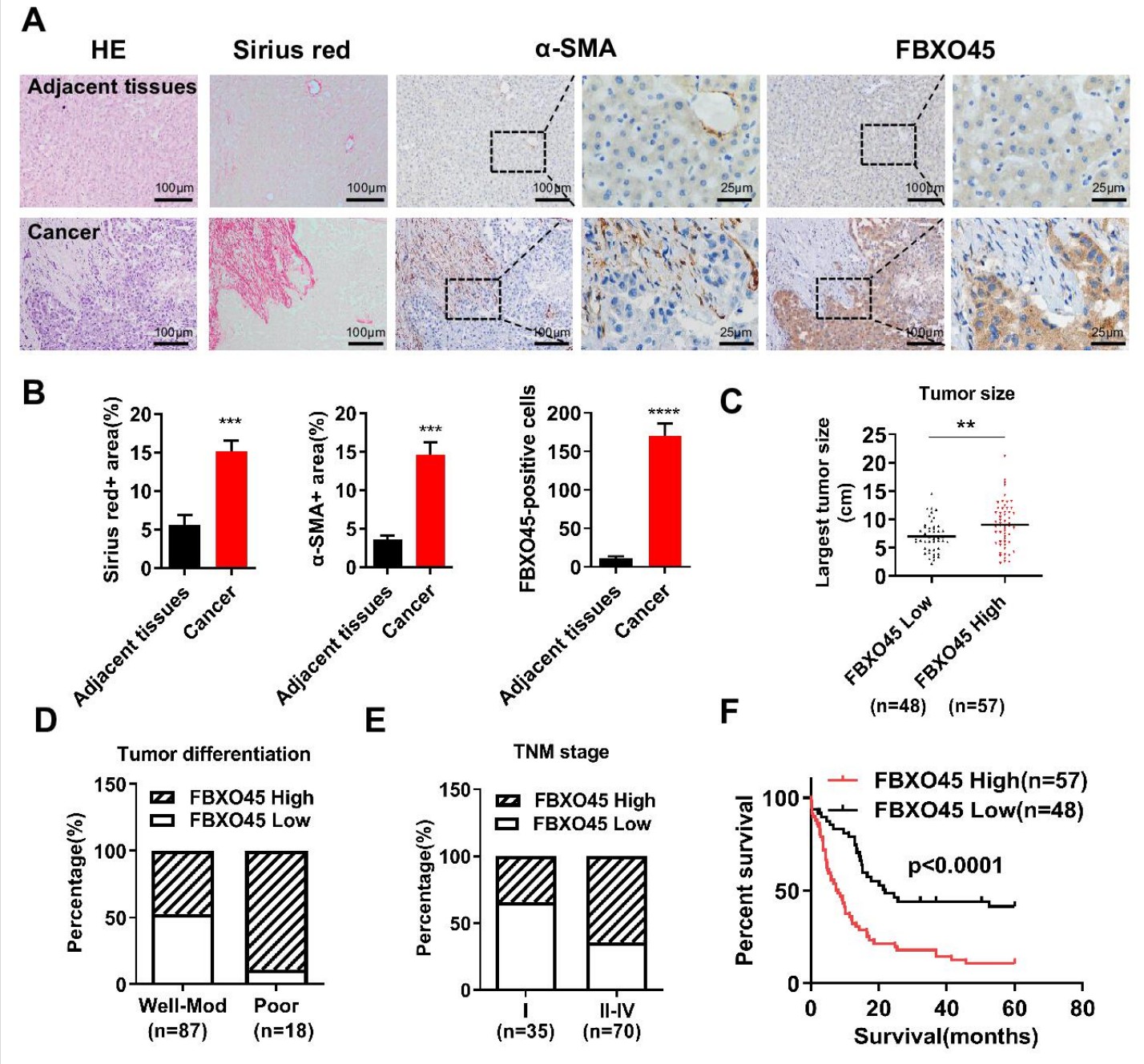

**Figure 1.** FBXO45 is associated with poor survival in HCC patients. (**A, B**) Tissue sections of HCC and matched adjacent tissues from patients were subjected to HE staining, Sirius red staining, or IHC staining for α-SMA and FBXO45. Representative images are shown (**A**). The proportions of Sirius red-positive, α-SMA-positive, and FBXO45-positive areas were quantified (**B**). Data are represented as the mean± SEM, n=5, ***p≤0.001, ****p≤0.0001. (**C–E**) The relationships between FBXO45 protein expression and tumor size (**C**), tumor differentiation (**D**), and TNM stage (**E**) were evaluated. (**F**) The association between the FBXO45 protein expression and overall survival of HCC patients was determined. Data are represented as the mean± SEM, **p≤0.01. HCC, hepatocellular carcinoma.*Figure 1—source data 1* for A; *Figure 1—source data 2* for B.

The online version of this article includes the following figure supplement(s) for figure 1:

**Source data 1.** Clinical and pathological data from HCC patient.

**Source data 2.** FBXO45 is associated with poor survival in HCC patients.

**Figure supplement 1.** *FBXO45* mRNA level is associated with poor survival in HCC patients.

**Table 1.** Relationships between the FBXO45 protein and clinicopathologic characteristics in 105 HCC patients.

| Variables | Cases | FBXO45 High level | FBXO45 Low level | P value |
|---|---|---|---|---|
| Age(years) | | | | 0.582 |
| ≤55 | 76 | 40 | 36 | |
| >55 | 29 | 17 | 12 | |
| Gender | | | | 0.864 |
| Female | 16 | 9 | 7 | |
| Male | 89 | 48 | 41 | |
| TNM stage | | | | 0.004** |
| I | 35 | 12 | 23 | |
| II–IV | 70 | 45 | 25 | |
| Histologic grade | | | | 0.001** |
| G1G2 | 87 | 41 | 46 | |
| G3 | 18 | 16 | 2 | |
| Tumor size | | | | 0.239 |
| ≤5 cm | 23 | 10 | 13 | |
| >5 cm | 82 | 47 | 35 | |
| Recurrence | | | | 0.631 |
| Present | 24 | 12 | 12 | |
| Absent | 81 | 45 | 36 | |
| Metastasis | | | | 0.098 |
| Present | 53 | 33 | 20 | |
| Absent | 52 | 24 | 28 | |

Calculated using the $x^2$ test.
**$p≤0.01$ were considered statistically significant.

of FBXO45 localized to HCC cells but not hepatic stellate cells (HSCs; identified by alpha-smooth muscle actin [α-SMA], a well-established marker of HSCs in the fibrotic liver) or fibrotic cells (Sirius red signal, an indicator of fibrotic cells), implying that the expression of FBXO45 was restricted to HCC cells (*Figure 1A–B*). Strikingly, gene expression analysis of two data sets revealed that *FBXO45* was overexpressed in HCC tissue compared with normal tissue (*Figure 1—figure supplement 1A*; *Chen et al., 2002*; *Wurmbach et al., 2007*).

The relationships between FBXO45 protein expression and clinicopathological parameters were investigated. IHC analysis on HCC samples showed that FBXO45 expression was positively associated with tumor size, tumor differentiation, and TNM stage (*Figure 1C–E* and *Table 1*), indicating that FBXO45 expression positively correlated with HCC disease progression. To extend this work, the associations between overall survival (OS) and various risk factors in 105 HCC tissue samples were analyzed. Univariate and multivariate analyses demonstrated that FBXO45 (p<0.001), tumor differentiation (p=0.021), and metastasis status (p=0.002) were significantly associated with OS in HCC patients, suggesting that FBXO45 is an independent risk factor for poor OS in HCC patients (OS, hazard ratio [HR]: 2.447; 95% confidence interval [CI]: 1.490–4.021; p<0.001, *Supplementary file 1*). Kaplan-Meier analysis demonstrated that FBXO45 expression in HCC patients was positively correlated with poor survival (p<0.0001; *Figure 1F*). Strikingly, data from The Cancer Genome Atlas (TCGA) database showed that the expression of *FBXO45* was significantly correlated with AFP, tumor differentiation, tumor stage, and TNM stage (*Figure 1—figure supplement 1B-D* and *Supplementary file 2*). Furthermore, *FBXO45* was an independent prognostic marker in HCC (p=0.041; *Supplementary file 3*), and high *FBXO45* expression was positively related to poor survival in HCC patients (p=0.03; *Figure 1—figure supplement 1E*).

## FBXO45 promotes fibrosis, inflammation, and hepatocarcinogenesis in mice

Subsequently, the role of FBXO45 in hepatocarcinogenesis in vivo was addressed. To achieve this, a mouse model with FBXO45 hyperexpression (FBXO45-OE) was generated and injected intraperitoneally with DEN/CCl$_4$ to promote hepatocarcinogenesis and shorten the observation period. DNA and protein were isolated from liver tissues of FBXO45-OE and wild-type (WT) mice. The transgenic status was confirmed by PCR analysis (*Figure 2—figure supplement 1A*) and Western blot analysis (*Figure 2—figure supplement 1B-C*). The high FBXO45 expression in FBXO45-OE mice (n=11) resulted in more tumors in the liver than the low FBXO45 expression in WT mice (n=20) at up to 24 weeks post-injection, indicating that high FBXO45 expression drives hepatocarcinogenesis in mice (*Figure 2A–B*). Furthermore, compared with WT mice, FBXO45-OE mice developed a high liver:body weight ratio (*Figure 2C*). In line with this finding, the cell proliferation markers *Mki67*, proliferating cell

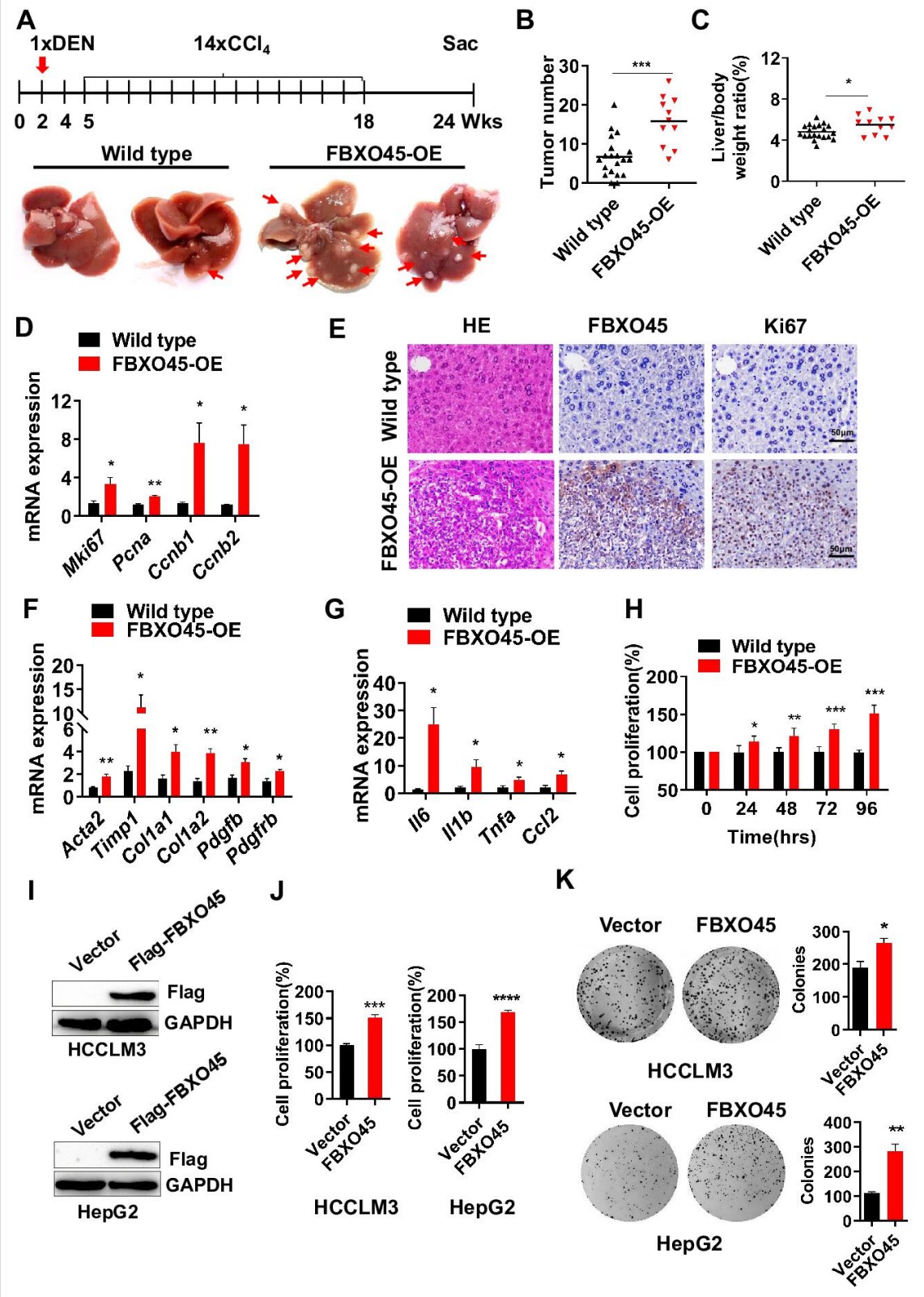

**Figure 2.** FBXO45 promotes hepatocarcinogenesis in mice. (**A–C**) FBXO45-OE (n=11) and wild-type (n=20) male mice were injected with DEN at the age of 2 weeks, followed by 14 injections of CCl₄; the mice were sacrificed at 24 weeks of age. Representative images (**A**), tumor number (**B**), and the liver/body weight ratio (**C**) are shown. Arrows indicate tumors; *p≤0.05, ***p≤0.001. (**D–G**) Mice were treated as described for (**A–C**). The transcription of proliferation-related genes (*Mki67*, *Pcna*, *Ccnb1*, and *Ccnb2*) (**D**), fibrotic genes (*Acta2*, *Timp1*, *Col1a1*, *Col1a2*, *Pdgfb*, and *Pdgfrb*) (**F**), and inflammatory

*Figure 2 continued on next page*

*Figure 2 continued*

markers (*Il6*, *Il1b*, *Tnfa*, and *Ccl2*) (**G**) in whole-liver tissues was assessed by qPCR. An IHC staining assay with the indicated antibodies was shown (**E**). Data are represented as the mean± SEM, n=6, *p≤0.05, **p≤0.01, ***p≤0.001. (**H**) The primary cells were isolated from different tumors from different mice. FBXO45-OE primary hepatocytes grew faster than wild-type cells in a time-dependent manner. Data are represented as the mean± SEM, n=5, *p≤0.05, **p≤0.01, ***p≤0.001. (**I–K**) HCCLM3 and HepG2 cells were transfected with an empty vector or Flag-FBXO45 plasmids for 48 hr. A portion of cells were reseeded into 96-well plates and cell proliferation was analyzed by a CCK-8 assay after 48 h. (**J**) The other portion of cells were plated into six-well plates and colonies counted after 9–13 days. (**K**) Data are represented as the mean± SEM, n=6 (**J**), n=3 (**K**). *p≤0.05, **p≤0.01, ***p≤0.001, ****p≤0.0001. *Figure 2—source data 1* for B, C, D, F, G, H, J and K.

The online version of this article includes the following figure supplement(s) for figure 2:

**Source data 1.** FBXO45 promotes hepatocarcinogenesis in mice.

**Figure supplement 1.** FBXO45 silencing inhibits the proliferation of HCC cells.

**Figure supplement 1—source data 1.** FBXO45 silencing inhibits the proliferation of HCC cells.

nuclear antigen (*Pcna*), cyclin B1 (*Ccnb1*), and cyclin B2 (*Ccnb2*) were significantly upregulated in the FBXO45-OE mice compared with the WT mice (*Figure 2D–E*).

Fibrosis and inflammation develop in response to hepatic injury and are associated with the development of HCC (*Liang et al., 2019*; *Ruart et al., 2019*). Next, whether FBXO45-driven hepatocarcinogenesis is accompanied by hepatic fibrosis and inflammation was evaluated by qPCR. The results showed that the transcription of the predominant fibrotic genes, including *Acta2*, tumor inhibitor of metalloproteinase 1 (*Timp1*), collagen type 1 alpha 1 (*Col1a1*), collagen type 1 alpha 2 (*Col1a2*), platelet-derived growth factor β (*Pdgfb*), and platelet-derived growth factor receptor β (*Pdgfrb*), was significantly increased in the FBXO45-OE mice (*Figure 2F*). Furthermore, the expression of inflammatory markers, including interleukin *Il6*, *Il1b*, tumor necrosis factor α (*Tnfa*), and monocyte chemoattractant protein 1 (*Ccl2*), was dramatically increased in the FBXO45-OE mice (*Figure 2G*). Thus, hepatic fibrosis and inflammation were involved in the development of HCC triggered by FBXO45.

To confirm the oncogenic role of FBXO45 in vivo, the function of FBXO45 in vitro was evaluated. Compared with WT cells, primary FBXO45-OE hepatocytes grew significantly faster (*Figure 2H*). In line with this finding, the expression of the proliferation markers *Mki67*, *Pcna*, and *Ccnb1* was upregulated in FBXO45-OE cells (*Figure 2—figure supplement 1D*). Also, overexpression of FBXO45 promoted cell proliferation and colony formation in human HCC cells (*Figure 2I–K*), while silencing of FBXO45 inhibited these capacities (*Figure 2—figure supplement 1E-G*).

## FBXO45 promotes polyubiquitination at the Lys190 and Lys450 sites and subsequent activation of IGF2BP1

To identify the direct substrates of FBXO45, human HCC HCCLM3 cells were transfected with Flag-tagged FBXO45. Cell lysates were precipitated with an anti-Flag M2 IgG or a normal nontargeting IgG, followed by mass spectrometry (MS) to analyze the interacting proteins. The ubiquitinated proteins in FBXO45-overexpressing tumors in comparison to normal tissues were analyzed by MS. Thus, the interacting proteins and ubiquitinated proteins were compared. Six interacting proteins had more than 1.5-fold changes in ubiquitinated protein level in the Flag-FBXO45 group compared with the control group (*Figure 3A* and *Supplementary file 4*). IGF2BP1 was one of the most upregulated oncogenes and played a critical role in the induction of HCC tumorigenesis in previous studies (*Gutschner et al., 2014*).

To further validate the immunoprecipitation (IP)-MS results, IP and Western blot analysis were used to analyze the binding between FBXO45 and IGF2BP1. FBXO45, upon ectopic expression in HCCLM3 or HepG2 cells, pulled down endogenous IGF2BP1 (*Figure 3B*). Reciprocally, endogenous IGF2BP1 pulled down ectopically expressed FBXO45 (*Figure 3C*). Significantly, WT FBXO45, but not its F-box domain-deleted mutant(FBXO45ΔF), pulled down IGF2BP1 in both HepG2 and HCCLM3 cells (*Figure 3D*), while both WT FBXO45 and FBXO45ΔF pulled down N-cadherin (a FBXO45 target protein binding the Sprouty domain) (*Chung et al., 2014*; *Figure 3—figure supplement 1A*), suggesting that IGF2BP1 interacted with FBXO45 at the F-box domain. In addition, FBXO45 and IGF2BP1 were co-localized in cytoplasm, which further supports the binding of these two proteins (*Figure 3—figure supplement 1B*). Based on the analysis of the ubiquitome shown in *Figure 3A*, we found that IGF2BP1 in HCC exhibited the upregulated ubiquitination at Lys190 and Lys450. However, it is still

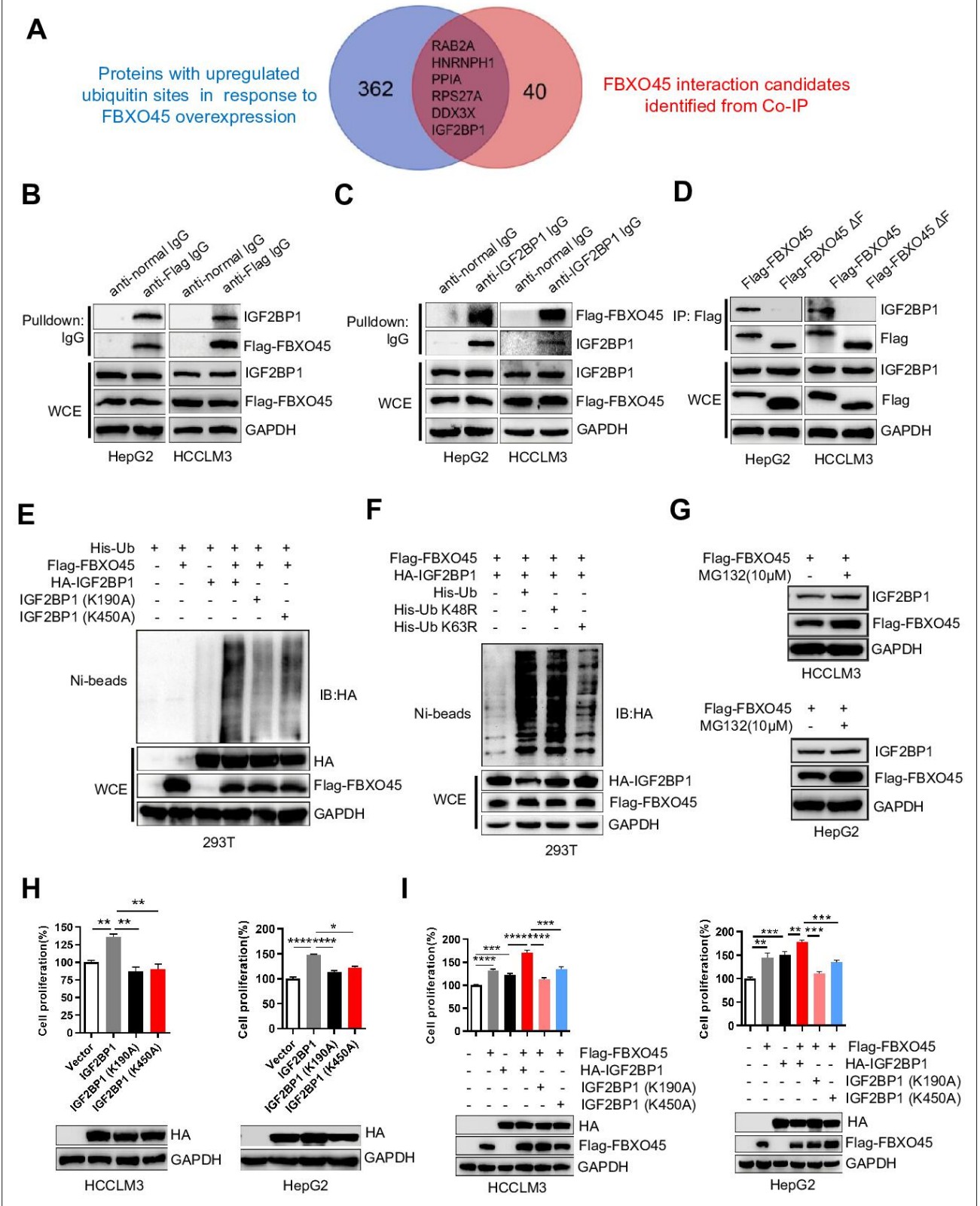

**Figure 3.** FBXO45 promotes polyubiquitination at the Lys190 and Lys450 sites and subsequent activation of IGF2BP1. (**A**) The Venn diagram shows the number of proteins with upregulated ubiquitin sites identified in response to FBXO45 overexpression (blue), FBXO45 interaction candidates identified by Co-IP and mass spectrometry (red), and overlapping proteins in the datasets. (**B, C**) FBXO45 bound to endogenous IGF2BP1. HepG2 and HCCLM3 cells were transfected with Flag-FBXO45 plasmids for 48 hr and then lysed. The cell lysates were added to the indicated antibodies and Protein A/G

*Figure 3 continued on next page*

*Figure 3 continued*

PLUS-Agarose, followed by Western blot analysis. WCE, whole cell extract. (**D**) HepG2 and HCCLM3 cells were transfected with Flag-FBXO45 or Flag-FBXO45 ΔF for 48 hr, followed by immunoprecipitation with anti-Flag antibody and Western blot analysis. (**E**) IGF2BP1 was polyubiquitylated by FBXO45, and K190 or K450 mutation partly blocked this effect. HEK293T cells were transfected with the indicated plasmids, followed by pull-down using Ni-NTA beads or direct Western blot analysis with the indicated antibodies. (**F**) FBXO45 mediated K63-linked polyubiquitination of IGF2BP1. HEK293T cells were transfected with the FBXO45 and IGF2BP1 plasmids along with WT-ubiquitin or its mutants (K63R and K48R), followed by pull-down with Ni-NTA beads or direct Western blot analysis with the indicated antibodies. (**G**) HepG2 and HCCLM3 cells were transfected with Flag-FBXO45 for 45 hr and then treated with or without MG132 (10 μM) for 3 hr. Blot shows the expression levels of IGF2BP1. (**H**) HepG2 and HCCLM3 cells were transfected with wild-type, K190-mutant or K450-mutant IGF2BP1 and subjected to Western blot analysis and a CCK-8 assay. Data are represented as the mean± SEM, n=3 (HCCLM3), n=6 (HepG2), *p≤0.05, **p≤0.01, ****p ≤0.0001. (**I**) HepG2 and HCCLM3 cells were co-transfected with indicated plasmids for 48 hr, followed by Western blot analysis or CCK-8 assay. Data are represented as the mean± SEM, n=6 (HCCLM3), n=5 (HepG2), **p≤0.01, ***p≤0.001, ****p≤0.0001. Co-IP, coimmunoprecipitation. *Figure 3—source data 1* for A; *Figure 3—source data 2* for H and I.

The online version of this article includes the following figure supplement(s) for figure 3:

**Source data 1.** Ubiquitinated proteins and interacted proteins with FBXO45.

**Source data 2.** FBXO45 promotes polyubiquitination at the Lys190 and Lys450 sites and subsequent activation of IGF2BP1.

**Figure supplement 1.** Two sites K190 and K450 in IGF2BP1 are conserved.

**Figure supplement 2.** FBXO45 promotes IGF2BP1 polyubiquitination via PAM.

unclear whether the oncogenic activity of IGF2BP1 is related to its ubiquitination. Here, these two sites were evolutionarily conserved among different species (*Figure 3—figure supplement 1C*). Next, whether FBXO45 promoted IGF2BP1 ubiquitination was evaluated. As shown in *Figure 3E*, FBXO45 enhanced IGF2BP1 polyubiquitination, while the K190A or K450A mutants significantly blocked this action. To characterize the type of ubiquitin chains involved in FBXO45-mediated IGF2BP1 polyubiquitination, two ubiquitin mutants (K48R and K63R) were used along with the WT ubiquitin. The in vivo ubiquitination assay indicated that a K63R ubiquitin mutant significantly attenuated the formation of polyubiquitin chain to IGF2BP1 compared with both WT and a K48R mutant form of ubiquitin (*Figure 3F*). In line with this finding, K48 ubiquitin attenuated IGF2BP1 polyubiquitination compared with K63 and WT ubiquitin (*Figure 3—figure supplement 1D*), which further confirmed that FBXO45 promoted K63-linked polyubiquitination of IGF2BP1. FBXO45 formed a complex with the E3 ubiquitin ligase PAM (*Richter et al., 2020*; *Saiga et al., 2009*). In order to know whether the ubiquitylation of IGF2BP1 was dependent on PAM activity, ubiquitination assay was performed to investigate the effect of PAM knockdown on IGF2BP ubiquitination. As shown in *Figure 3—figure supplement 2A*, knockdown of PAM could significantly block FBXO45-mediated IGF2BP1 ubiquitination. Furthermore, IGF2BP1 interacted with SKP1 but not cullin-1 (*Figure 3—figure supplement 2B*). Taken together, FBXO45-PAM-SKP1 promoted K63-linked ubiquitination of IGF2BP1.

In addition, MG132, an inhibitor of proteasome, did not significantly increase IGF2BP1 protein level compared with FBXO45-overexpressed alone group, indicating that FBXO45 doesn't mediate IGF2BP1 proteasome-dependent degradation in HCCs (*Figure 3G*). In order to validate whether FBXO45 increased IGF2BP1 activity via polyubiquitination at K190 and K450, the effect of FBXO45 and its mutants on cell proliferation was analyzed by CCK-8 assay. WT IGF2BP1 dramatically promoted HCC proliferation, whereas K190A and K450A mutants attenuated its oncogenic activity (*Figure 3H*). In line with this finding, FBXO45 promoted WT IGF2BP1-mediated cell proliferation much stronger than two IGF2BP1 mutants (K190A and K450A) (*Figure 3I*). Taken together, the ubiquitination of IGF2BP1 at K190 and K450 was positively related to its oncogenic activity, which was mediated by FBXO45.

## FBXO45 promotes cell proliferation via IGF2BP1-PLK1 axis

The RNA-binding protein IGF2BP1 is an important protumorigenic factor in liver carcinogenesis (*Gutschner et al., 2014*). Consistently, IGF2BP1 significantly promoted cell proliferation and colony formation when overexpressed, while IGF2BP1 silencing inhibited cell proliferation and colony formation (*Figure 4A–B*). How does IGF2BP1 promote HCC cell proliferation? Simon et al. reported that IGF2BP1 may control several genes, including *PLK1*, *MKI67*, *FOXK1*, and *PDLIM7* (*Müller et al., 2019*). Here, IGF2BP1 silencing significantly inhibited the expression of PLK1 and KI67, two regulators of cell proliferation, at mRNA and protein levels in both HepG2 and HCCLM3 cells, in which *MKI67* transcript was reported to be stabilized by IGF2BP1 previously (*Figure 4C–D*; *Gutschner et al., 2014*).

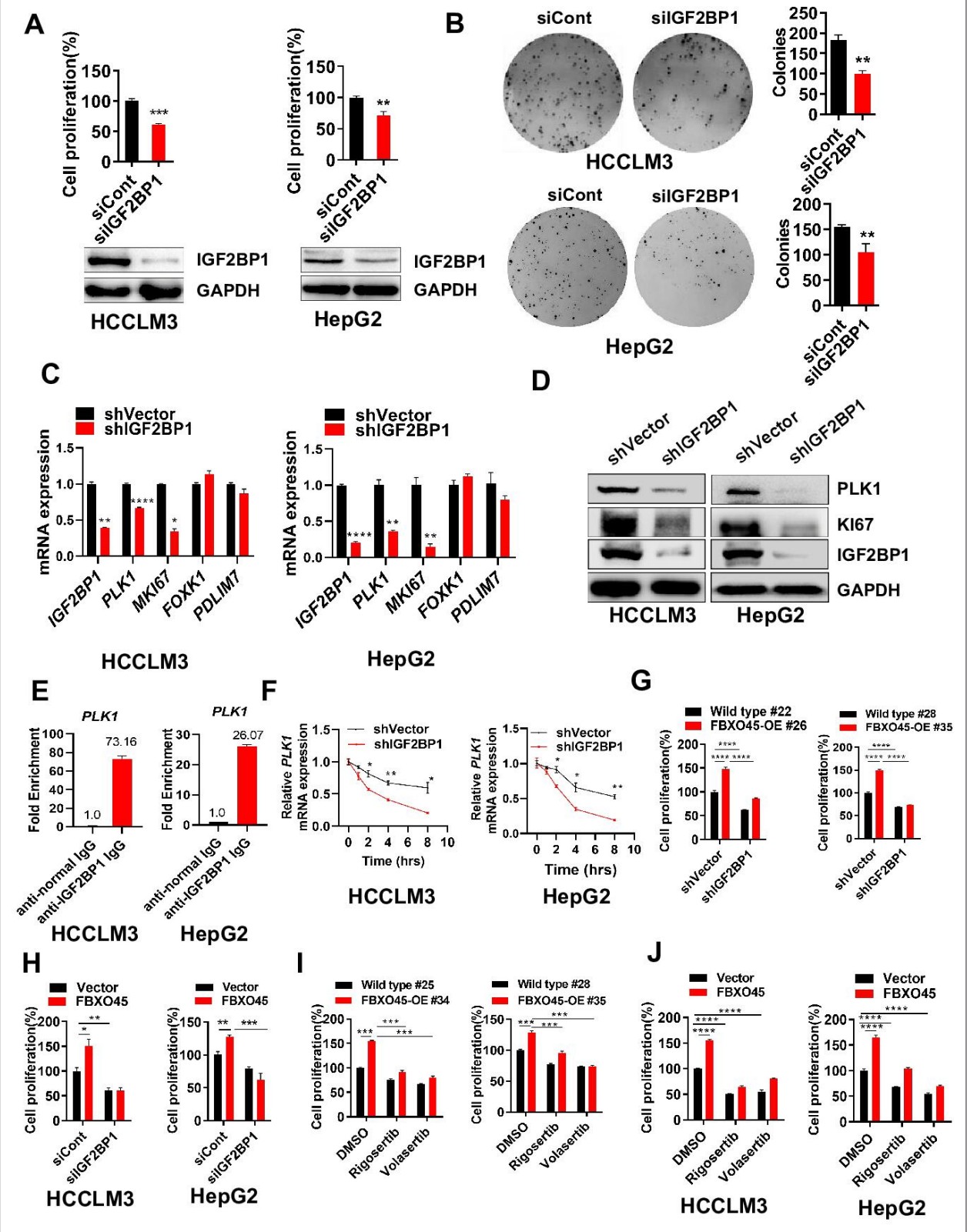

**Figure 4.** FBXO45 promotes cell proliferation via IGF2BP1-PLK1 axis. (**A, B**) HCCLM3 and HepG2 cells were transfected with siControl or siRNA targeting IGF2BP1 (siIGF2BP1) for 48 h. A portion of cells were reseeded into 96-well plates and cell proliferation analyzed by CCK-8 assay after 48 h (**A**). The other portion of cells were plated into six-well plates and colonies counted after 9–13 days (**B**). (**C, D**) Cells were transfected with shVector or shIGF2BP1. Gene and protein expression were analyzed by qPCR (**C**) and Western blot analysis (**D**), respectively. (**E**) RIP-PCR assay showed that IGF2BP1

*Figure 4 continued on next page*

*Figure 4 continued*

bound *PLK1* mRNA in HepG2 and HCCLM3 cells. Cells were added beads with anti-normal IgG or anti-IGF2BP1 IgG antibody. The purified *PLK1* mRNA was analyzed by RT-PCR. (**F**) HCCLM3 and HepG2 cells were transfected with shVector or shIGF2BP1 and then treated with 10 µg/ml actinomycin D for different time intervals, followed by qPCR (**G**) The primary cells were isolated from different tumors from different mice. FBXO45-OE and wild-type (WT) primary hepatocytes were transfected with the indicated shVector or shIGF2BP1, followed by analysis of cell proliferation using a CCK-8 assay. (**H**) HCCLM3 and HepG2 cells were transfected with Flag-FBXO45 alone or in combination with siIGF2BP1 for 48 hr, followed by a CCK-8 assay. (**I**) FBXO45-OE and WT primary hepatocytes were isolated from different tumors from different mice and treated with PLK1 inhibitors Rigosertib or Volasertib (2 µM) for 48 hr, followed by analysis of cell proliferation using a CCK-8 assay. (**J**) HCCLM3 and HepG2 cells were transfected with empty vector or Flag-FBXO45 for 24 hr and then exposed to Rigosertib or Volasertib (0.5 µM) for 24 hr, and subjected to a CCK-8 assay. Data are represented as the mean± SEM, n=3 (**B, C, E, F**), n=4 (A, H-HepG2), n=6 (H-HCCLM3), (**G, I, J**), *p≤0.05, **p≤0.01, ***p≤0.001, ****p≤0.0001. *Figure 4—source data 1* for A, B, C, E, F, G, H, I and J.

The online version of this article includes the following figure supplement(s) for figure 4:

**Source data 1.** FBXO45 promotes cell proliferation via IGF2BP1-PLK1 axis.

**Figure supplement 1.** IGF2BP1binds and stabilizes *PLK1* mRNA.

**Figure supplement 1—source data 1.** IGF2BP1binds and stabilizes *PLK1* mRNA.

RIP-PCR analysis showed that IGF2BP1 interacted with *PLK1* mRNA in both HepG2 and HCCLM3 cells (*Figure 4E* and *Figure 4—figure supplement 1A*). Furthermore, inhibition of RNA polymerase by actinomycin D (act D) significantly reduced the levels of *PLK1* and *PEG10* mRNA (*PEG10* is a well-known target mRNA stabilized by IGF2BP1; *Zhang et al., 2021*) and silencing of IGF2BP1 further enhanced this action (*Figure 4F* and *Figure 4—figure supplement 1B*). Taken together, IGF2BP1 binds *PLK1* mRNA and regulates PLK1 expression via promoting *PLK1* mRNA stability.

Whether FBXO45-mediated cell proliferation could be attributed to IGF2BP1 was subsequently, evaluated by CCK-8 assay. In primary hepatocytes, FBXO45 promoted cell proliferation and this effect was dramatically attenuated by IGF2BP1 silencing (*Figure 4G*). In line with this finding, silencing of IGF2BP1 also significantly blocked FBXO45-mediated cell proliferation in HepG2 and HCCLM3 cells (*Figure 4H*). To further validate FBXO45-IGF2BP1-PLK1 axis, HCCLM3, HepG2 cells, or primary hepatocytes were treated with Rigosertib and Volasertib (the new selective inhibitors of PLK1). The data showed that these two drugs could significantly block FBXO45-mediated cell proliferation, indicating that PLK1 is the downstream target of FBXO45 (*Figure 4I–J*). Taken together, FBXO45 promotes cell proliferation via IGF2BP1-PLK1 axis.

## IGF2BP1 knockdown protects against FBXO45-mediated hepatocarcinogenesis in vivo

To examine whether targeting IGF2BP1 suppresses FBXO45-driven hepatocarcinogenesis in vivo, FBXO45-OE mice were injected intraperitoneally with a recombinant adeno-associated virus nine carrying an IGF2BP1-specific shRNA with liver-restricted expression (AAV9-shIGF2BP1-eGFP-(TBG)) at the age of 10 days. At the age of 2 weeks, the mice were then injected intraperitoneally with DEN/CCl₄. The animals were euthanized at the age of 24 weeks. Whether IGF2BP1 silencing could suppress FBXO45-driven hepatocarcinogenesis in mice was preliminarily evaluated by analyzing liver tumor number and the liver:body weight ratio. The images showed that the IGF2BP1-silenced group developed fewer neoplasms than the control group (*Figure 5A–B*), indicating that interfering IGF2BP1 blocked FBXO45-driven hepatocarcinogenesis. The liver:body weight ratio was not significantly different between the two groups (*Figure 5C*). Furthermore, cell proliferation-related biomarkers PLK1 and Ki67 were reduced in IGF2BP1-silenced tissues compared with the control group (*Figure 5D*). IHC staining consistently demonstrated that silencing of IGF2BP1 reduced tumorigenesis and the levels of the downstream proteins PLK1 and Ki67 expression (*Figure 5E*). Thus, therapeutic targeting of IGF2BP1 may offer options for intervention in FBXO45-triggered HCC.

## PLK1 inhibition protects against FBXO45-mediated HCC xenograft tumors in vivo

To evaluate whether targeting PLK1 suppresses FBXO45-driven HCC xenograft tumors in vivo, HCCLM3-FBXO45 cells (4×10⁶) suspended in PBS were mixed with Matrigel and injected subcutaneously into the right flank of nude mice and allowed to grow. When tumors grew to a volume

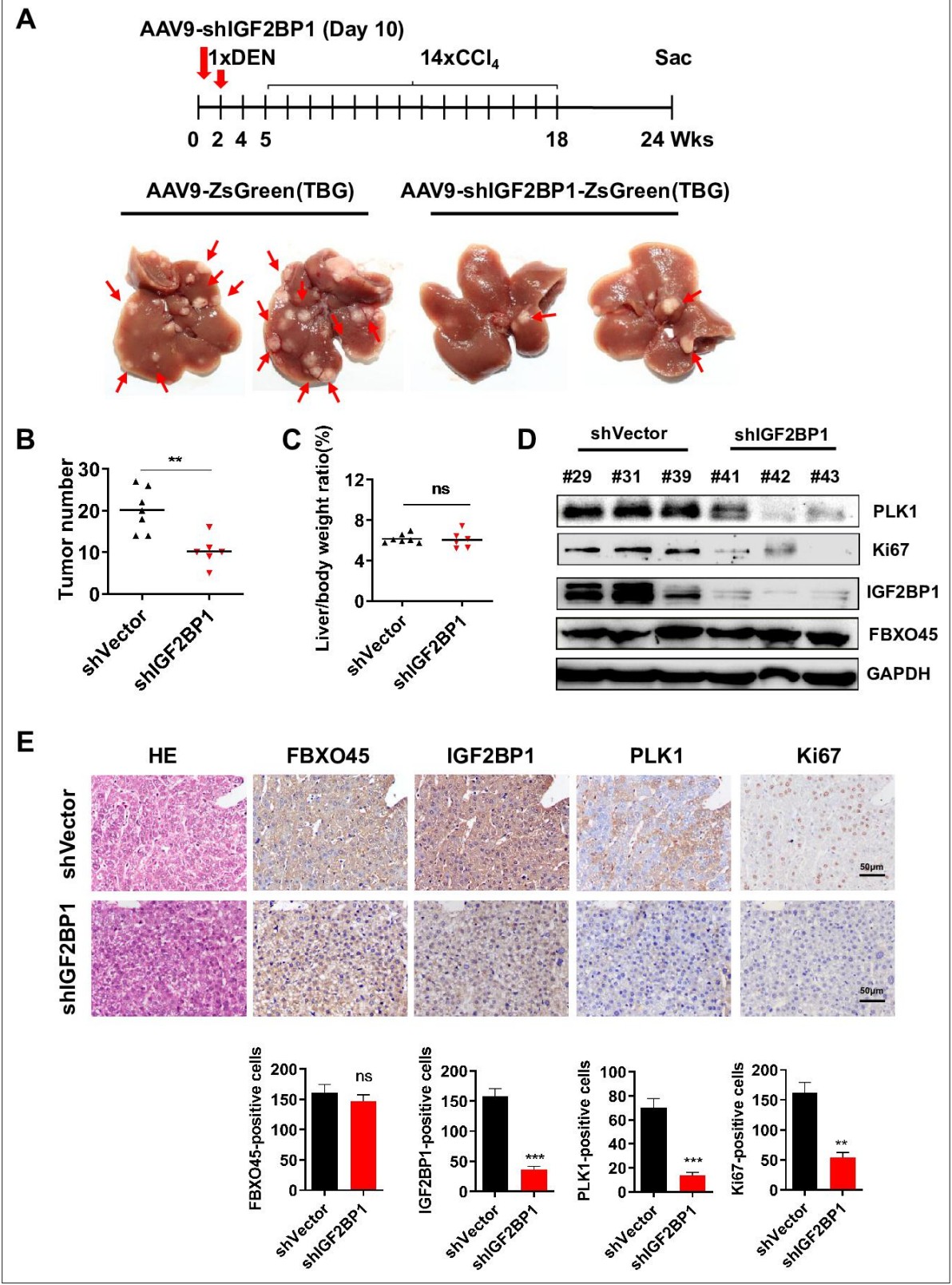

**Figure 5.** Interfering IGF2BP1 protects against FBXO45-mediated hepatocarcinogenesis. (**A–E**) FBXO45-OE male mice were injected with AAV9-shIGF2BP1-ZsGreen(TBG) ($5 \times 10^{10}$ gene copies per mouse, i.p.) (n=6) or AAV9-ZsGreen(TBG) (n=7) at the age of 10 days, followed by injection of DEN (25 mg/kg, i.p.) at the age of 2 weeks, and then 14 weekly injections of $CCl_4$ (0.5 ml/kg, i.p.) to shorten observation periods. The mice were sacrificed 22 weeks after DEN administration. Representative images (**A**), tumor number (**B**), and the liver/body weight ratio (**C**) are shown. Livers were collected

*Figure 5 continued on next page*

*Figure 5 continued*

and followed by Western blot analysis (**D**) or IHC staining (**E**) with the indicated antibodies. Data are represented as the mean± SEM (**B, C, E**), n=4 (**E**), ns, nonsignificant, \*\*p≤0.01, \*\*\*p≤0.001. *Figure 5—source data 1* for B, C and E.

The online version of this article includes the following figure supplement(s) for figure 5:

**Source data 1.** Interfering IGF2BP1 protects against FBXO45-mediated hepatocarcinogenesis.

of approximately 80–120 mm³, the animals were received the 15 mg/kg volasertib treatment (volasertib was suspended in saline with 40% PEG-400, 5% Tween-80). After 2 weeks, the animals were euthanized. Whether PLK1 inhibition could suppress FBXO45-driven HCC xenograft tumors in mice was preliminarily evaluated by analyzing body weight and tumor volume. The images showed that the PLK1 inhibition group grew smaller neoplasms than the control group (*Figure 6A–B*). The body weight was not significantly different between the four groups (*Figure 6C*). Furthermore, IHC staining consistently demonstrated that inhibition of PLK1 reduced tumorigenesis and the levels of the downstream CCNB2 and Ki67 expression (*Figure 6D*).

## Correlation between the FBXO45, IGF2BP1, and PLK1 levels in HCC tissue

The IGF2BP1 expression in 105 paired HCC samples was examined by immunohistochemistry. Immunohistochemistry analysis revealed that the IGF2BP1 protein was highly expressed in the tumor tissue samples compared with the adjacent normal tissue samples (*Figure 7A*). Furthermore, high expression of IGF2BP1 (log-rank test, p=0.0015) or PLK1 (log-rank test, p=0.0035) were positively associated with poor OS in HCC patients (*Figure 7B*). Subsequently, the relationship between IGF2BP1 expression and clinicopathological characteristics was evaluated. High IGF2BP1 expression was significantly associated with poor clinicopathological characteristics, including tumor size, advanced TNM stage, and poor differentiation (*Figure 7C–E* and *Supplementary file 5*), indicating that IGF2BP1 may have a critical function in HCC. The correlation between FBXO45, IGF2BP1, and PLK1 was finally evaluated by IHC staining. FBXO45 staining was positively correlated with IGF2BP1 staining (70/105) in the 105 paired HCC tissue samples, of which 45.7% (48/105) had high expression of both proteins. PLK1 staining was positively correlated with IGF2BP1 staining (70/105) in the 105 paired HCC tissue samples, of which 50.5% (53/105) had high expression of both proteins (*Figure 7F–G*). Furthermore, the HCC patients with high expression of FBXO45, IGF2BP1, and PLK1 had shorter OS time compared with those patients with low expression of these proteins (*Figure 7—figure supplement 1A*). Consistent with this finding, high expression of FBXO45, IGF2BP1, and PLK1 was positively associated with tumor size, TNM stage, and tumor poor differentiation (*Figure 7—figure supplement 1B-D*), indicating that FBXO45-IGF2BP1-PLK1 axis positively correlated with HCC disease progression. This finding suggested that FBXO45 may promote IGF2BP1 activation and upregulate PLK1 in HCC. Taken together, the elevated expression of FBXO45 in the liver of mice or patients promoted hepatocarcinogenesis via induction of IGF2BP1 ubiquitination and activation and PLK1 upregulation (*Figure 7H*).

## Discussion

HCC has poor treatment options and a poor prognosis due to the unclear process of hepatocarcinogenesis. Research evidence has indicated a close link between abnormal expression of E3 ubiquitin ligases and HCC (*Shen et al., 2018*; *Li et al., 2014*). Recent studies have reported that FBXO45 may play key roles in the tumorigenesis and prognosis of squamous cell lung carcinoma and colorectal cancer (*Wang et al., 2018*; *Wu et al., 2019*). However, it is still unknown whether FBXO45 promotes HCC tumorigenesis in vivo and the underlying mechanism is also unclear. Here, the data showed that (1) FBXO45 functioned as an oncogene by promoting HCC tumorigenesis in vivo. (2) FBXO45 protein expression in HCC tissue was increased and significantly associated with poor survival in HCC patients. (3) Multivariate analysis revealed that FBXO45 was an independent risk factor for patient survival. (4) FBXO45 promoted HCC tumorigenesis via IGF2BP1 polyubiquitination and activation. (5) Targeting IGF2BP1-PLK1 axis protected against HCC tumorigenesis triggered by elevated expression of FBXO45 in vitro and in vivo. Our findings elucidated the oncogenic role of FBXO45 in HCC

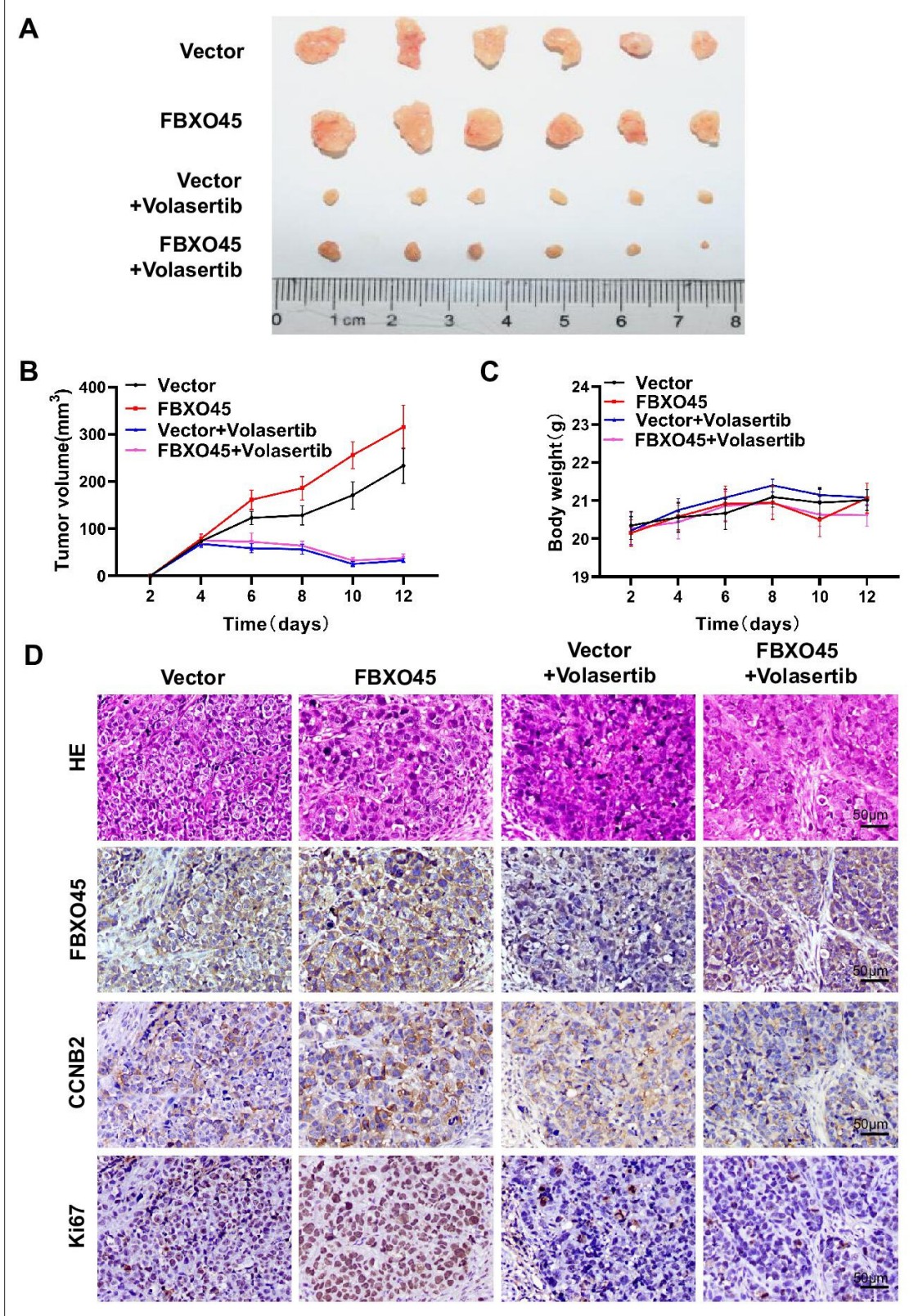

**Figure 6.** Inhibiting PLK1 blocks FBXO45-mediated the growth of HCC xenograft tumors in mice. (**A–D**) A total of $4 \times 10^6$ HCCLM3-FBXO45 cells were injected subcutaneously into right flank sides of nude mice. When the tumors reached a size of 80–120 mm$^3$, the mice were received i.p. injection of volasertib at dose of 15 mg/kg every 2 days for 8 days. After 8 days, mice were euthanized and tumors were photographed (**A**). The tumor volumes were monitored and growth curves were plotted (**B**). The body weight of the mice was also measured and plotted (**C**). Proliferation-related biomarkers

*Figure 6 continued on next page*

*Figure 6 continued*

Ki67 and CCNB2 were stained in tumors by IHC staining (**D**). Shown are mean± SEM, n=6. HCC, hepatocellular carcinoma. *Figure 6—source data 1* for B and C.

The online version of this article includes the following figure supplement(s) for figure 6:

**Source data 1.** Inhibiting PLK1 blocks FBXO45-mediated the growth of HCC xenograft tumors in mice.

---

tumorigenesis as well as the mechanism underlying this process. Targeting IGF2BP1-PLK1 axis might represent a novel therapeutic strategy for the treatment of human HCC with high FBXO45 expression.

In addition to the role of FBXO45 in the regulation of emotional expression (*Shimanoe et al., 2019*), recent studies have demonstrated that FBXO45 binds specifically to the tumor suppressor PAR4 (*Chen et al., 2014*), transcription factor p73 (a member of the p53 family) (*Peschiaroli et al., 2009*), and FBXW7 (*Richter et al., 2020*), triggering ubiquitin-mediated proteolysis and regulating cell survival. Conversely, elevated expression of FBXO45 results in decreased proteolysis of N-cadherin and promotion of neuronal differentiation (*Chung et al., 2014*). In contrast to these previous studies, these genes were not involved in FBXO45-triggered HCC, as indicated by analysis of the ubiquitin sites and FBXO45-binding proteins (*Figure 3A*). Our data suggested that FBXO45 specifically bound to IGF2BP1 and promoted its polyubiquitination and activation in HCC. IGF2BP1, an oncogene, was one of the strongly upregulated RNA-binding proteins among 116 proteins identified in HCC tissue compared with normal liver tissue (*Gutschner et al., 2014*). At the molecular level, IGF2BP1 bound to and stabilized the mRNA transcripts and increased the protein expression of MYC and KI67, two potent regulators of cell proliferation and apoptosis (*Gutschner et al., 2014*). In addition, several IGF2BP1 target genes were identified, including *PDLIM7*, F*OXK1*, *PLK1*, and *ATG5*, some of which were validated in previous studies (*Müller et al., 2019*). Here, IGF2BP1 significantly upregulated the transcription of PLK1 and KI67 (two proteins related to cell proliferation) compared with that of other genes in HCCLM3 and HepG2 cells. Furthermore, FBXO45 increased the expression of these two proteins, and silencing IGF2BP1 blocked this effect in vitro and in vivo, indicating the promotive effect of the FBXO45-IGF2BP1-PLK1 axis on hepatocarcinogenesis.

IGF2BP1 is a known essential regulator of intestinal development and cancer. *Chatterji et al., 2019*. reported increased IGF2BP1 expression in patients with Crohn's disease or ulcerative colitis. In addition, recent studies demonstrated that IGF2BP1, an oncogene, was overexpressed in many cancers, especially HCC, and promoted cell growth, cell proliferation, or metastasis (*Liu et al., 2018*; *Elcheva et al., 2019*; *Ghoshal et al., 2019*; *Müller et al., 2018*). However, it remains unknown whether the oncogenic activity of IGF2BP1 is related to its ubiquitination. Here, an ubiquitome study indicated that FBXO45 promoted IGF2BP1 polyubiquitination at K190 and K450. An ubiquitination assay demonstrated that mutation of K190 or K450 could significantly block the FBXO45-mediated polyubiquitination of IGF2BP1, reduce IGF2BP1 activity and inhibit cell proliferation. Therefore, the ubiquitination of IGF2BP1 at K190 and K450 is positively related to its activity in hepatocarcinogenesis.

In summary, FBXO45 played an important role in promoting hepatocarcinogenesis in vitro and in vivo. There was a positive correlation between increased FBXO45 and IGF2BP1 expression in HCC tissue, and these two proteins were associated with poor survival in patients with HCC. Targeting IGF2BP1-PLK1 axis may be a novel approach for treating HCC exhibiting high FBXO45 expression.

## Materials and methods
### Human subjects

In total, 105 human HCC and paired adjacent paraffin tissue specimens were obtained from the Department of Hepatobiliary Surgery, Southwest Hospital, Chongqing, PR China. The tissue samples were used for immunohistochemistry(IHC) analysis. This study was approved by the Ethics Committee of Southwest Hospital (KY2020127), and all of the patients provided informed consent. Detailed clinical and pathological data were obtained from each patient (See *Figure 1—source data 1*).

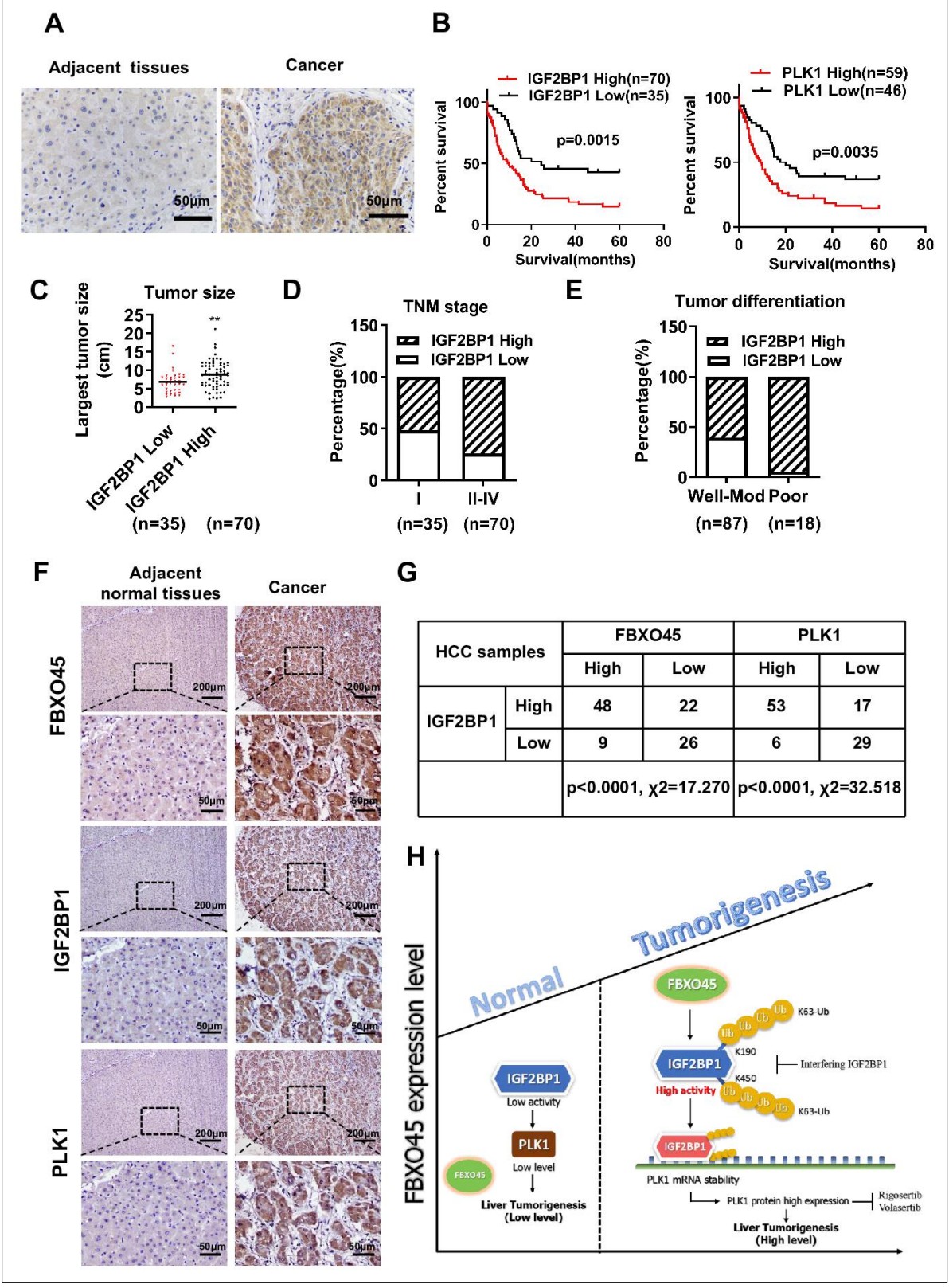

**Figure 7.** Correlation between the FBXO45, IGF2BP1, and PLK1 levels in HCC tissues. (**A**) The expression of IGF2BP1 in HCC tissue samples was determined by staining with an anti-IGF2BP1 antibody. Scale bar, 50 μm. (**B**) The association between IGF2BP1 or PLK1 protein expression and overall survival in HCC patients was evaluated. (**C–E**) The relationships between the IGF2BP1 protein and tumor size (**C**), TNM stage (**D**), and differentiation (**E**) were determined. **p≤0.01. (**F, G**) The expression of IGF2BP1, PLK1, and FBXO45 in HCC tissue samples was determined by IHC staining.

*Figure 7 continued on next page*

*Figure 7 continued*

Representative images of stained tumors are shown (**F**). The association between FBXO45, IGF2BP1, and PLK1 was statistically significant (**G**). Scale bar, 200 µm. (**H**) A model of hepatocarcinogenesis triggered by elevated FBXO45 expression driven via IGF2BP1-PLK1 axis. HCC, hepatocellular carcinoma.

The online version of this article includes the following figure supplement(s) for figure 7:

**Figure supplement 1.** Relationships between the high or low expression of both FBXO45/IGF2BP1/PLK1 and clinicopathologic characteristics in HCC patients.

## Animal studies and a DEN/CCl$_4$-induced HCC animal model

FBXO45-OE mice high-expressing *Fbxo45* allele were generated by Cyagen Biosciences Inc (Guangzhou). To generate this mouse strain, fertilized mouse zygotes were coinjected with a mixture of Cas9 mRNA, gRNA to mouse ROSA26 gene, and a construct containing mouse *Fbxo45* complementary DNA (cDNA) and then transferred into pseudopregnant C57BL/6N female mice. PCR and gene sequencing were used to identify targeted mice. Only male mice were used in this study. To further promote liver tumorigenesis, FBXO45-OE or WT mice were injected intraperitoneally with DEN (25 mg/kg, Sigma-Aldrich, N0756) at the age of 2 weeks, followed by 14 weekly injections of CCl$_4$ (0.5 ml/kg, dissolved in corn oil). At the age of 24 weeks, mice were euthanized. The tumor number was counted, and the largest tumor diameter was measured. Livers were photographed. The liver and body were weighed. Mouse breeding was performed in the specific pathogen-free facility of the Animal Center of Army Medical University. All animal experiments were approved by the Institutional Animal Care and Use Committee (IACUC) of Army Medical University and complied with all relevant ethical regulations (AMUWEC2020936).

## Isolation and culture of primary cells

Primary murine cells were isolated from the liver of FBXO45-OE mice or WT mice treated with DEN and CCl$_4$. The detailed cell isolation procedure was described previously (*Mederacke et al., 2015*). Briefly, first, cannulation of an anaesthetized mouse via the inferior vena cava was performed, and then the liver was stably perfused with 35 ml EGTA solutions (8 g/L NaCl, 0.4 g/L KCl, 88.17 mg/L NaH$_2$PO$_4$·H$_2$O, 120.45 mg/L Na$_2$HPO$_4$, 2.38 g/L HEPES, 0.35 g/L NaHCO$_3$, 0.19 g/L EGTA, and 0.9 g/L Glucose; pH 7.35–7.4) until the liver was swollen and pale in colour, 35 ml pronase solutions (8 g/L NaCl, 0.4 g/L KCl, 88.17 mg/L NaH$_2$PO$_4$·H$_2$O, 120.45 mg/L Na$_2$HPO$_4$, 2.38 g/L HEPES, 0.35 g/L NaHCO$_3$, 0.56 g/L CaCl$_2$·2H$_2$O, and 0.4 g/L pronase; pH 7.35–7.4) for 5 min and 40 ml collagenase solutions (8 g/L NaCl, 0.4 g/L KCl, 88.17 mg/L NaH$_2$PO$_4$·H$_2$O, 120.45 mg/L Na$_2$HPO$_4$, 2.38 g/L HEPES, 0.35 g/L NaHCO$_3$, 0.56 g/L CaCl$_2$·2H$_2$O, and 92.5 U collagenase; pH 7.35–7.4) for 7 min. After in situ digestion, the liver was carefully removed and minced. The minced liver was further digested with 50 ml pronase/collagenase solution (8 g/L NaCl, 0.4 g/L KCl, 88.17 mg/L NaH$_2$PO$_4$·H$_2$O, 120.45 mg/L Na$_2$HPO$_4$, 2.38 g/L HEPES, 0.35 g/L NaHCO$_3$, 0.56 g/L CaCl$_2$·2H$_2$O, 0.5 g/L pronase, 88 U collagenase, and 10 ml DNase I; pH 7.35–7.4) at 40°C for 20 min. The liver cell suspension was filtered through a 70-µm cell strainer to remove undigested debris. Cells were centrifuged at 580×*g* for 10 min at 4°C, and the cells were washed two times with GBSS/B (8 g/L NaCl, 0.37 g/L KCl, 0.21 g/L MgCl$_2$·6H$_2$O, 70 mg/L MgSO$_4$·7H$_2$O, 59.6 mg/L Na$_2$HPO$_4$, 30 mg/L KH$_2$PO$_4$, 0.991 g/L glucose, 0.227 g/L NaHCO$_3$, and 0.225 g/L CaCl$_2$·2H$_2$O; pH 7.35–7.4). Finally, the primary cells were cultured in Dulbecco's modified Eagle's medium (DMEM) supplemented with 20% fetal bovine serum (FBS) and 1% penicillin and streptomycin.

## Cell culture studies

The human HCC cell line HCCLM3 was obtained from BeiNa Culture Collection (China, Beijing), the Huh7 cell line was obtained from the Fudan Cell Bank (China, Shanghai), and the HepG2 cell line was obtained from ATCC. All cell lines tested negative for mycoplasma contamination. All experiments were carried out in DMEM containing 10% FBS, 100 U/ml penicillin, and 100 mg/ml streptomycin (Invitrogen, USA) at 37°C and 5% CO$_2$.

## qRT-PCR

Total RNA was isolated from whole-liver samples or cultured cells using RNAiso Plus reagent (TaKaRa, Japan), and cDNAs were generated using the PrimeScript RT Reagent Kit with gDNA Eraser (Perfect

**Table 2.** Primers used for qRT-PCR.

| | Gene symbol | Primer forward | Primer reverse |
|---|---|---|---|
| Human | IGF2BP1 | CTCCTTTATGCAGGCTCCCG | GGGTCTCCAGCTTCACTTCC |
| | PLK1 | CGAGTTCTTTACTTCTGGCT | TATTGAGGACTGTGAGGGGC |
| | MKI67 | CTTCGCTCTTACTCCCCTGC | GACTGAGACCACAGGACTGC |
| | FOXK1 | CCAAGGATGAGTCAAAGCCG | GTTCAAAGAGAGGTTGTGCC |
| | PDLIM7 | CTCACAGGCACCGAGTTCAT | CACTGTGCTCGTTTTGTCCG |
| | PEG10 | GCCTAGAAATGGGCCGTTGT | CGTTCTTGTCGTTGGTGAAC |
| Mouse | Acta2 | CCGCCATGTATGTGGCTATT | CAGTTGTACGTCCAGAGGCATA |
| | Timp1 | CTCGGACCTGGATGCTAAAA | ACTCTTCACTGCGGTTCTGG |
| | Col1a1 | TAAGGGTCCCCAATGGTGAGA | GGGTCCCTCGACTCCTACAT |
| | Col1a2 | CCAGCGAAGAACTCATACAGC | GGACACCCCTTCTACGTTGT |
| | Pdgfb | GGTGAGCAAGGTTGTAATGG | GGAGGCAATGGACAGACAA |
| | Pdgfrb | TCCCACATTCCTTGCCCTTC | GCACAGGGTCCACGTAGATG |
| | Mki67 | AAAGGCGAAGTGGAGCTTCT | TTTCGCAACTTTCGTTTGTG |
| | Pcna | AAAGATGCCGTCGGGTGAAT | CCATTGCCAAGCTCTCCACT |
| | Ccnb1 | AGCGAAGAGCTACAGGCAAG | CTCAGGCTCAGCAAGTTCCA |
| | Ccnb2 | CCGACGGTGTCCAGTGATTT | AGGTTTCTTCGCCACCTGAG |
| | Tnfa | CAAGATGCTGGGACAGTGAC | AGGGAAGAATCTGGAAAGGT |
| | Ccl2 | AAAAACCTGGATCGGAACCAA | CGGGTCAACTTCACATTCAAAG |
| | Il6 | GAGCCCACCAAGAACGATAG | TCATTTCCACGATTTCCCAG |
| | Il1b | GCTTCAGGCAGGCAGTATCA | GACAGCACGAGGCTTTTTTG |
| | Fbxo45 | GCTGGGAAGTGACGACCAGAG | AGCCAGAAGTCAGATGCTCAAGG |

Real Time) (TaKaRa, Japan) according to the manufacturer's instructions. Then, the mRNA levels of the indicated genes were analyzed by qRT-PCR using TB Green Premix Ex Taq II (TaKaRa, Japan) and the primers listed in *Table 2* on the CFX96 Touch Real-time PCR Detection System. Results were calculated based on the threshold cycle (Ct), and the relative fold change was determined using the $2^{-\Delta\Delta Ct}$ method.

For actinomycin D (Sigma-Aldrich, USA) experiments, cells were transfected with shIGF2BP1 before the addition of 10 µg/ml actinomycin D in their culture media for different time periods.

## Plasmids, siRNA, lentivirus, and an adeno-associated virus

The human *FBXO45* gene tagged with 3xFlag and human *IGF2BP1* tagged with HA were cloned into a pcDNA3.1+ vector for exogenous expression. The human *FBXO45 ΔF* gene tagged with 3xFlag and deleted 39–82 amino acid was cloned into a pcDNA3.1+ vector for exogenous expression. We utilized a site-directed mutagenesis technique to substitute lysine residues with alanine at the 190 and 450 sites of IGF2BP1 by using the QuikChange Lightning Site-Directed Mutagenesis Kit (Agilent, USA). siRNAs against FBXO45 and IGF2BP1 and a nontargeting siRNA were synthesized by GenePharma (China, Shanghai), and the sequences are shown in *Table 3*. The adeno-associated virus AAV9-shIGF2BP1-ZsGreen-TBG with liver-restricted expression was purchased from Hanheng Biosciences (China, Shanghai).

## Cell proliferation and colony formation assays

HCC cells were seeded in 96-well plates at 3000 cells per well for 24 hr and then transiently transfected with the indicated plasmids or siRNAs. After 48 hr, cell proliferation was analyzed using a Cell Counting Kit-8 (CCK-8; Dojindo, Japan) assay. Cell proliferation was expressed as relative cell growth

**Table 3.** RNA interference.

| | Gene symbol | Forward | Reverse |
|---|---|---|---|
| Human | FBXO45 1# | CCAGGAAUGUCUACAUUAAdTdT | UUAAUGUAGACAUUCCUGGdTdT |
| | FBXO45 2# | CCAGCAGUUUCUGCUGUAUdTdT | AUACAGCAGAAACUGCUGGdTdT |
| | FBXO45 3# | CAGAUAGGAGAAAGAAUUCGA | UCGAAUUCUUUCUCCUAUCUG |
| | IGF2BP1 2# | CCACCAGUUGGAGAACCAUdTdT | AUGGUUCUCCAACUGGUGGdTdT |
| | IGF2BP1 4# | CCGGGAGCAGACCAGGCAAdTdT | UUGCCUGGUCUGCUCCCGGdTdT |
| | MYCBP2 #1 | CCCGAGAUCUUGGGAAUAAUU | UUAUUCCCAAGAUCUCGGGUU |
| | No-targeting control | UUCUCCGAACGUGUCACGUUU | ACGUGACACGUUCGGAGAAUU |
| Mouse | shIGF2BP1 | 5'-GATCCGCCGGGAGCAGACCAGGCAAT TCAAGAGATTGCCTGGTCTGCTCC CGGTTTTTTACGCGTG-3' | |

in percentage, which was compared with the control group. We set the control group as 100. For colony formation assays, 500 treated cells were plated in six-well plates in 1 ml DMEM containing 10% FBS and 1% penicillin and streptomycin. During colony growth, the culture medium was replaced every 3 days. Colonies >50 cells were counted 14 days after plating.

## Western blot analysis

Treated cells were homogenized in RIPA lysis buffer (Thermo Fisher Scientific, USA) supplemented with a protease inhibitor mixture (Roche, Switzerland) and then incubated on ice for 20 min. After incubation, the lysates were centrifuged at 13,000 rpm for 15 min at 4°C. After centrifugation, the supernatant was collected into a new Eppendorf (EP) tube. A BCA Protein Assay Reagent Kit (Beyotime, China) was used to quantify the concentration of protein. After denaturing, samples were separated by SDS-PAGE (Beyotime, China) and then transferred to NC membranes (GE Healthcare, UK). The following antibodies were used for Western blot analysis: rabbit anti-IGF2BP1 (1:1000), rabbit anti-PLK1 (1:1000), anti-Cullin-1 antibody (1:1000), goat anti-mouse IgG HRP (1:5000), goat anti-rat IgG HRP (1:5000), and goat anti-rabbit IgG HRP (1:5000) antibodies were purchased from Cell Signaling Techology (Cell Signaling, Danvers, MA). A mouse anti-Flag antibody (1:1000) was purchased from Sigma-Aldrich (St. Louis, MO). A rabbit anti-FBXO45 antibody (1:1000) was obtained from Abcam (Cambridge, MA). Anti-SKP1 antibody and anti-GAPDH antibody (1:5000) were obtained from Proteintech (China, Wuhan). A rat anti-HA antibody (1:1000) was purchased from Roche (Basel, Switzerland). The antibodies used in this study are listed in Appendix 1.

## In vivo ubiquitination assay

To examine the ubiquitination of IGF2BP1 by FBXO45, 293T cells were transfected with His-Ub and Flag-FBXO45 with HA-IGF2BP1, HA-IGF2BP1 (K190A), or HA-IGF2BP1 (K450A). After 72 hr of transfection, the cells were treated with 10 μM MG132 for 4–6 hr before harvest for an in vivo ubiquitination assay. The in vivo ubiquitination assay was performed as described previously. In brief, the cells were lysed in buffer A (6 mol/L guanidinium-HCl, 0.1 mol/L $Na_2HPO_4/NaH_2PO_4$, 10 mmol/L Tris-HCl [pH 8.0], 5 mmol/L imidazole and 10 mmol/L β-mercaptoethanol) and exposed to 50 μl Ni-NTA beads (Qiagen, Valencia, CA) overnight. The beads were washed with buffer A plus 10 mM β-mercaptoethanol, buffer B (8 mM urea, 0.1 M $Na_2HPO_4/NaH_2PO_4$, 10 mM Tris/HCl [pH 8.0], and 10 mM β-mercaptoethanol), buffer C (8 mM urea, 0.1 M $Na_2HPO_4/NaH_2PO_4$, 10 mM Tris/HCl [pH 6.3], 10 mM β-mercaptoethanol, and 0.2% Triton X-100), and buffer C plus 10 mM β-mercaptoethanol and 0.1% Triton X-100. His6-tagged ubiquitinated proteins were then eluted with buffer D (200 mM imidazole, 0.15 M Tris-HCl (pH 6.7), 30% glycerol, 0.72 M β-mercaptoethanol, and 5% SDS). The elution was analyzed by Western blot analysis with an anti-HA antibody.

## Immunohistochemistry staining assay

Tissue specimens were fixed in 10% formalin overnight and embedded in paraffin. The paraffin-embedded HCC tissue samples were cut into 4-μm-thick tissue sections, and then the tissue sections

were mechanically deparaffinized and incubated in a sodium citrate antigen retrieval solution (Solarbio, USA) at high pressure for 2.5 min. After endogenous peroxidase activity was blocked with methanol containing 3% hydrogen peroxide for 15 min, the tissue sections were incubated with primary antibodies at 4°C overnight, followed by incubation with a secondary antibody at room temperature for 60 min. Then, the sections were stained with DAB (ZSGB-BIO, China). Subsequently, the sections were counterstained with haematoxylin (Biosharp, China) for 20 s. The following primary antibodies were used: anti-α-SMA (Abcam, USA), anti-FBXO45 (Bioss, China), anti-IGF2BP1 (Proteintech, China), anti-PLK1 (Boster, China), and anti-Ki67 (Millipore, USA) antibodies. The staining was evaluated by different specialized pathologists without any knowledge of the patient characteristics. The staining intensity was determined using Spot Denso function of AlphaEaseFC software. The positive stained cells were counted out of a total of 500 cells on average from three different areas from three patients for each group. The antibodies used in this study are listed in Appendix 1.

## Coimmunoprecipitation assay

To examine the direct interaction between FBXO45 and IGF2BP1, HCCLM3 and HepG2 cells were transfected with Flag-FBXO45. After 48 hr of transfection, the cells were lysed in an immunoprecipitation lysate buffer (20 mM Tris-HCl, pH 8.0, 100 mM NaCl, 1% NP-40, and a protease inhibitor cocktail tablet) for 20 min on ice. The lysates were centrifuged at $10,000 \times g$ for 10 min at 4°C. After centrifugation, the supernatant was collected into a new EP tube. The cell lysates were added to 1–10 µl (0.2–2 µg) primary antibody (the optimal antibody concentration was determined by titration) and incubated for 1 h at 4 °C. After incubation, 50 µl Protein A/G PLUS-Agarose (Santa Cruz, CA) was added to the protein-antibody complexes and incubated at 4°C on a rotating device overnight. The immunoprecipitates were washed four times with immunoprecipitation buffer, and a 2× sample loading buffer was added to the beads before boiling for 5 min. The supernatant was collected and used in a Western blot assay. The antibodies used in this study are listed in Appendix 1.

## Proteome and ubiquitome analysis

Cell Samples were sonicated three times on ice using a high intensity ultrasonic processor in lysis buffer (8 M urea, 1% Protease Inhibitor Cocktail). The remaining debris was removed by centrifugation at $12,000 \times g$ at 4°C for 10 min. Then the supernatant was collected and the protein concentration was determined with BCA Kit according to the manufacturer's instructions. For digestion, the protein was reduced with 5 mM dithiothreitol at 56°C for 30 min, followed by alkylated with 11 mM iodoacetamide at room temperature in darkness for 15 min. The protein sample was then diluted by adding 100 mM TEAB to urea concentration less than 2 M. Finally, trypsin was added at 1:50 trypsin-to-protein mass ratio for the first digestion overnight and 1:100 trypsin-to-protein mass ratio for a second 4-h digestion. The samples were subsequently subjected to LC-MS/MS analysis. The tryptic peptides were dissolved in 0.1% formic acid (solvent A), directly loaded onto a home-made reversed-phase analytical column (15 cm length, 75 µm i.d.). The gradient was comprised of an increase from 6% to 23% solvent B (0.1% formic acid in 98% acetonitrile) over 26 min, 23% to 35% in 8 min and climbing to 80% in 3 min then holding at 80% for the last 3 min, all at a constant flow rate of 400 nl/min on an EASY-nLC 1000 UPLC system. The peptides were subjected to NSI source followed by tandem mass spectrometry (MS/MS) in Q ExactiveTM Plus (Thermo Fisher Scientific) coupled online to the UPLC. The electrospray voltage applied was 2.0 kV. The m/z scan range was 350–1800 for full scan, and intact peptides were detected in the Orbitrap at a resolution of 70,000. Peptides were then selected for MS/MS using NCE setting as 28 and the fragments were detected in the Orbitrap at a resolution of 17,500. A data-dependent procedure that alternated between one MS scan followed by 20 MS/MS scans with 15.0 s dynamic exclusion. Automatic gain control was set at 5E4. Fixed first mass was set as 100 m/z. 1.2-fold-of-change and p-value<0.05 was defined as statistically significant for the proteome and 1.5-fold-of-change for ubiquitome.

## RNA-binding protein immunoprecipitation assay

Magna RIP RNA-Binding Protein Immunoprecipitation Kit (Millipore, USA) was used according to the manufacturer's protocol. In order to detect the *PLK1* mRNA that binds to IGF2BP1 protein. HCCLM3 and HepG2 cells were lysed in 100 µl RIP lysis buffer (containing 0.5 µl Protease Inhibitor Cocktail and 0.25 µl RNAse inhibitor) on ice for 5 min. The supernatant was collected by centrifugation. A portion

of supernatant was as the input. The other portion of supernatant was added 50 μl beads complex with 5 μg normal lgG or anti-IGF2BP1 antibody, and incubated overnight at 4°C. Then, beads were washed. Purification of RNA species were directly measured using quantitative reverse transcription polymerase chain reaction (RT-PCR).

## Immunofluorescence assay

Cultured cells were grown on coverslips placed in a six-well plate. Forty-eight hours after transfection, cells were subsequently fixed by 4% paraformaldehyde and permeabilized by Triton X-100. After blockage, cells were incubated with primary antibodies at 4°C overnight and secondary antibodies with fluorophore label at room temperature for 1 hr, in moist environmental box. At last, Hoechst 33342 was added to stain nucleus of cells. Cell images were captured using a Nikon ECLIPSE Ti microscopy.

## Tumor xenograft model

To assess in vivo tumor growth, 24 male BALB/c athymic nude mice, aged 6 weeks, were purchased from the animal centre of Army Medical University and provided with sterile water and food. The mice were separated into four groups by simple randomization, and each experimental group consisted of six mice (n=6). HCCLM3-FBXO45 cells ($4 \times 10^6$) suspended in PBS were mixed with Matrigel injected subcutaneously into the right flank of nude mice and allowed to grow. When the tumors reached a size of 80–120 mm$^3$, the mice were received volasertib (suspended in saline with 40% PEG-400, 5% Tween-80) via intraperitoneal (i.p.) injection at the dose of 15 mg/kg every 2 days for 8 days. Tumor volumes and body weight were monitored every other day. After 8 days, the mice were euthanized. To ensure death, cervical dislocation was used. Tumors were harvested and weighed. Tumor volume was calculated with the following formula: volume=π/6× length × width$^2$.

## Statistics

Prism 8.0 software (GraphPad, San Diego, CA) was used to analyze data, which are presented as the mean± SEM. Comparisons between two groups were performed with a two-tailed unpaired t-test. When more than two groups were compared, one-way analysis of variance with Tukey post hoc test was used when data passed Shapiro-Wilk normality test. When data did not pass normality test, nonparametric Kruskal-Wallis test was used followed by Dunn's Multiple Range test for post hoc comparisons. Area determination was performed using an imaging system (Olympus, Hamburg, Germany). Clinicopathological correlations were analyzed by Pearson's chi-square test. OS was analyzed using the Kaplan-Meier method and log-rank tests. The expression correlation between FBXO45 and IGF2BP1 was determined by Pearson's correlation coefficient. p-values≤0.05 were considered statistically significant.

## Acknowledgements

We would like to thank Yan Jiang, Yujun Zhang, Shu Chen, Jiejuan Lai, Ping Zheng and Ling Shuai for collection of tissues and clinical information for each patient.

## Additional information

### Funding

| Funder | Grant reference number | Author |
| --- | --- | --- |
| Third Military Medical University | 4174C6 | Chuan-Ming Xie |
| National Natural Science Foundation of China | 32071294 | Chuan-Ming Xie |

| Funder | Grant reference number | Author |
|---|---|---|

The funders had no role in study design, data collection and interpretation, or the decision to submit the work for publication.

## Author contributions

Xiao-Tong Lin, Data curation, Formal analysis, Investigation, Methodology, Validation, Visualization, Writing – original draft; Hong-Qiang Yu, Data curation, Formal analysis, Investigation, Methodology; Lei Fang, Investigation, Methodology; Ye Tan, Hao-Jun Xiong, Data curation, Investigation, Methodology; Ze-Yu Liu, Data curation, Methodology, Software, Visualization; Di Wu, Jie Zhang, Investigation, Methodology, Validation; Chuan-Ming Xie, Conceptualization, Data curation, Funding acquisition, Investigation, Project administration, Resources, Supervision, Validation, Writing – review and editing

## Author ORCIDs

Chuan-Ming Xie http://orcid.org/0000-0003-4362-6612

## Ethics

This study was approved by the Ethics Committee of Southwest Hospital. HCC samples were obtained from patients in Southwest Hospital after obtaining signed informed consent (KY2020127).
All animal experiments were approved by the Institutional Animal Care and Use Committee (IACUC) of Army Medical University and complied with all relevant ethical regulations (AMUWEC2020936).

## Decision letter and Author response

Decision letter https://doi.org/10.7554/eLife.70715.sa1

# Additional files

## Supplementary files

• Supplementary file 1. Univariate and multivariate analyses indicating associations between overall survival and various risk factors in 105 HCC patients.

• Supplementary file 2. Relationships between *FBXO45* mRNA expression and clinicopathologic characteristics in 253 HCC patients.

• Supplementary file 3. Univariate and multivariate analyses indicating the associations between overall survival and various risk factors in 253 HCC patients.

• Supplementary file 4. FBXO45-interacting proteins identified by Co IP-MS.

• Supplementary file 5. Relationships between the IGF2BP1 protein and clinicopathological characteristics in 105 HCC patients.

• Transparent reporting form

## Data availability

All data analysed during this study are included in the manuscript and supporting files. Source data files have been provided for Figures and Figure supplements contain images of gels or blots. Source data files have been provided for the original files of the gels or blots, named as "Source data 1 (blot images)". Source data files have been provided for the clinical and pathological data for HCC patients, named as "Figure 1-source data 1". Source data files have been provided for interacted proteins or ubiquitinated proteins, named as "Supplementary file 4. FBXO45-interacting proteins identified by Co IP-MS".

The following previously published datasets were used:

| Author(s) | Year | Dataset title | Dataset URL | Database and Identifier |
|---|---|---|---|---|
| Cancer Genome Atlas Research Network | 2013 | The Cancer Genome Atlas Pan-Cancer Analysis Project | https://portal.gdc.cancer.gov/projects/TCGA-LIHC | GDC Data Portal, TCGA-LIHC |

*Continued on next page*

*Continued*

| Author(s) | Year | Dataset title | Dataset URL | Database and Identifier |
|---|---|---|---|---|
| Wurmbach E, Chen YB, Khitrov G, Zhang W, Roayaie S, Schwartz M | 2007 | Genome-wide molecular profiles of HCV-induced dysplasia and hepatocellular carcinoma | https://www.ncbi.nlm.nih.gov/geo/query/acc.cgi?acc=GSE6764 | NCBI Gene Expression Omnibus, GSE6764 |
| Chen X, Cheung ST, So S, Fan ST, Barry C, Higgins J | 2002 | Gene expression patterns in human liver cancers | https://www.ncbi.nlm.nih.gov/geo/query/acc.cgi?acc=GSE3500 | NCBI Gene Expression Omnibus, GSE3500 |

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

# Appendix 1

## Appendix 1—key resources table

| Reagent type (species) or resource | Designation | Source or reference | Identifiers | Additional information |
|---|---|---|---|---|
| genetic reagent (*M. musculus*) | FBXO45-OE | Cyagen Biosciences Inc. | | *Fbxo45* knockin at the locus of ROSA26 in C57BL/6 mice |
| genetic reagent (*M. musculus*) | BALB/c athymic nude mice | Army Medical University | | Aged 6 weeks |
| cell line (*Homo-sapiens*) | Dermal fibroblast (normal, Adult) | ATCC | PCS-201–012 | |
| cell line (*Homo-sapiens*) | HepG2 (HCC,15-year-old Male) | ATCC | ATCC:HB-8065; RRID:CVCL_0027 | |
| cell line (*Homo-sapiens*) | Huh7 (HCC, Adult male) | Fudan Cell Bank | CSTR:19375.09.3101 HUMTCHu182; RRID:CVCL_0336 | |
| cell line (*Homo-sapiens*) | HCCLM3 | BeiNa Culture Collection | BNCC-342335; RRID:CVCL_6832 | |
| biological sample (*Macaca fascicularis*) | Primary simian hepatocytes | SingHealth, Singapore | | Freshly isolated from Macaca fascicularis |
| antibody | anti-IGF2BP1 (Rabbit Monoclonal) | Cell Signaling | Cat. #: 8,482 S; RRID:AB_11179079 | WB (1:1000) IP (1:150) |
| antibody | anti-IGF2BP1 (Rabbit Polyclonal) | Proteintech | Cat. #: 22803–1-AP; RRID:AB_2879173 | IHC (1:200) |
| antibody | anti-PLK1 (Rabbit Monoclonal) | Cell Signaling | Cat. #: 4,513 S; RRID:AB_2167409 | WB (1:1000) |
| antibody | anti-PLK1 (Mouse Monoclonal) | Boster | Cat. #: M00182 | IHC (1:300) |
| antibody | anti-Cullin-1 (Rabbit Polyclonal) | Cell Signaling | Cat. #: 4,995 S; RRID:AB_2261133 | WB (1:1000) |
| antibody | anti-FBXO45 (Rabbit Polyclonal) | Abcam | Cat. #:ab136614; | WB (1:1000) |
| antibody | anti-FBXO45 (Rabbit Polyclonal) | Bioss | Cat. #:bs-13150R; | IHC (1:300) |
| antibody | anti-$\alpha$-SMA (Rabbit Polyclonal) | Abcam | Cat. #:ab5694; RRID:AB_2223021 | IHC (1:1600) |
| antibody | anti-SKP1 (Rabbit Polyclonal) | Proteintech | Cat. #: 10990–2-AP; RRID:AB_2187492 | WB (1:1000) |
| antibody | anti-GAPDH (Rabbit Polyclonal) | Proteintech | Cat. #: 10494–1-AP; RRID:AB_2263076 | WB (1:5000) |
| antibody | anti-HA (Rat Monoclonal) | Roche | Cat. #: 11867423001; RRID:AB_390918 | WB (1:1000) |
| antibody | anti-N-cadherin (Mouse Monoclonal) | Cell Signaling | Cat. #: 14,215 S; RRID:AB_2798427 | WB (1:1000) |
| antibody | anti-Flag (Mouse Monoclonal) | Sigma-Baldric | Cat. #: F1804; RRID:AB_262044 | WB (1:1000) IF (1:200) |
| antibody | anti-CCNB2 (Rabbit Polyclonal) | Proteintech | Cat. #: 21644–1-AP; RRID:AB_10755304 | IHC (1:300) |
| antibody | Goat anti-mouse IgG HRP | Cell Signaling | Cat. #: 7,076 S; RRID:AB_330924 | WB (1:5000) |
| antibody | Goat anti-rabbit IgG HRP | Cell Signaling | Cat. #: 7,074 S; RRID:AB_2099233 | WB (1:5000) |
| antibody | Goat anti-rat IgG HRP | Cell Signaling | Cat. #: 7,077 S; RRID:AB_10694715 | WB (1:5000) |
| antibody | anti-Ki67 (Mouse Monoclonal) | Millipore | Cat#: FCMAB103AP; RRID:AB_10561769 | WB (1:1000) IHC (1:800) |
| antibody | Mouse IgG1 Isotype Control | Cell Signaling | Cat. #:5,415 S; RRID:AB_10829607 | |
| antibody | Rabbit IgG Isotype Control | Cell Signaling | Cat. #:3,900 S; RRID:AB_1550038 | |
| recombinant DNA reagent | pcDNA3.1-FBXO45-3xFLAG | This paper | | GenBank_ID:NM_001105573 |

*Appendix 1 Continued on next page*

*Appendix 1 Continued*

| Reagent type (species) or resource | Designation | Source or reference | Identifiers | Additional information |
|---|---|---|---|---|
| recombinant DNA reagent | pcDNA3.1-FBXO45ΔF-3xFLAG | This paper | | GenBank_ID:NM_001105573(del39-82aa) |
| recombinant DNA reagent | pcDNA3.1-IGF2BP1-HA | This paper | | GenBank_ID:NM_006546 |
| recombinant DNA reagent | AAV9-shIGF2BP1-ZsGreen-TBG | Hanheng Biosciences | | |
| recombinant DNA reagent | pLV[EXP]-EGFP-Puro-FBXO45-3xFLAG | This paper | | |
| recombinant DNA reagent | LV-U6-shIGF2BP1-mCherry-Puro | This paper | | |
| commercial assay or kit | Magna RIP RNA-Binding Protein Immunoprecipitation Kit | Millipore | Cat #:17–700 | |
| commercial assay or kit | PrimeScript RT reagent Kit with gDNA Eraser | Takara | Cat #: RR047 | |
| commercial assay or kit | TB Green Premix Ex Taq II | Takara | Cat #: RR820 | |
| commercial assay or kit | QuikChange Lightning Site-Directed Mutagenesis Kit | Agilent | Cat # : 210,519 | |
| chemical compound, drug | DNase I | Roche | Cat. #:10104159001 | |
| chemical compound, drug | Collagenase | Invitrogen | Cat. #:17104019 | |
| chemical compound, drug | DEN | Sigma-Aldrich | Cat. #: N0756 | 25 mg/kg |
| chemical compound, drug | Actinomycin D | Sigma-Aldrich | Cat. #: SBR00013 | 10 µg/mL |
| chemical compound, drug | MG-132, proteasome inhibitor | Selleck | Cat. #: S2619 | 10 µM |
| chemical compound, drug | Volasertib | Selleck | Cat. #: S2235 | |
| chemical compound, drug | Rigosertib | Selleck | Cat. #: S1362 | |
| chemical compound, drug | Protein A/G PLUS-Agarose | Santa Cruz | Cat. #: sc-2003; RRID:AB_10201400 | |
| Software, algorithm | Graphpad Prism | Graphpad | RRID:SCR_002798 | |
| Software, algorithm | IBM SPSS Statistics | IBM | RRID:SCR_019096 | |

