## [Decision Letter]

**Decision letter after peer review:**

Thank you for submitting your article "Elevated FBXO45 promotes liver tumorigenesis through enhancing IGF2BP1 ubiquitination and subsequent PLK1 upregulation" for consideration by *eLife*. Your article has been reviewed by 2 peer reviewers, and the evaluation has been overseen by a Reviewing Editor and Wafik El-Deiry as the Senior Editor. The reviewers have opted to remain anonymous.

Essential revisions:

The authors need to address the below raised concerns:

1. The overexpression of FBXO45 in the liver needs to be demonstrated at the protein level and evaluated in respect to impact independent of DEN/CCl4 treatment.

2. The analysis of liver tissue (Figure 2) need further specification. Were the analyses performed on tumor tissue or non-malignant liver tissue. How was tumor content considered?

3. The analyses of FBXO45 overexpression/depletion in liver cancer cell models remain puzzling. The DepMap portal clearly indicates a median positive essentiality score for FBXO45 in liver and other cancer cells (specifically in HepG2!). The authors report the opposite here based on overexpression and a single depletion without convincing control of depletion efficiency – barely any depletion is seen in Figure S2C. Here deletion studies and comparison with DepMap reported data is required.

4. The data on the ubiquitination of IGf2BP1 need further in detail evaluation and consideration. In Biogrid, ubiquitination of at least 35 lysine residues, incl. the here proposed have been reported. However, no association has been reported for FBXO45, but other E3 ubiquitin ligases. Although the data appear convincing, how can the data exclude additional/other lysine residues to be modified. Moreover, the authors need to evaluate the quantity of ubiquitinated IGF2BP1 to conclusively demonstrate if this modification could contribute substantially. Along these lines, the authors need to investigate how the modification of lysine residues influences IGF2BP1-controlled mRNA turnover/translation and/or RNA-association.

5. Although PLK1 likely is a target mRNA stabilized by IGF2BP1, the data presented in Figure 4C-E are only partially convincing. The steady state mRNA levels of PLK1 are only reduced by ~20%, whereas the protein content suggests a substantially more pronounced downregulation. Moreover, in one cell line FOXK1 is down- whereas it is upregulated in the other cell line upon IGF2BP1 depletion. Finally, the decay analyses (Figure 4E) require more time points, semi-logarithmic data presentation and the error bars are unlikely to result from biological but rather technical replicates. Additional positive and negative controls are required here. Finally, establishing IGF2BP1-dependent regulation of PLK1 expression requires investigation of IGF2BP1-PLK1 mRNA association and need to include studies on how IGF2BP1 depletion impairs PLK1-directed phosphorylation.

6. Figure 4F does not show a dramatic attenuation of FBXO45-promoted cell proliferation. Instead, it only demonstrates that IGF2BP1 depletion results in modestly reduced proliferation in both, wild type as well as FBXO45-overexpressing cells. How does depletion affect FBXO45 expression?

7. Analyses of PLK1 inhibition do not provide any useful insights. Here it needs to be demonstrated that IGF2BP1 deletion and FBXO45 overexpression modulate the potency of the respective inhibitors and that this is connected to IGF2BP1 ubiquitination.

8. Results presented in Figure 5 provide interesting hints for a role of IGF2BP1 in promoting HCC, as previously reported, and regulation of PLK1 expression, but they fail to demonstrate that this is linked to FBXO45. Why is FBXO45 expression not monitored here by Western blotting?

9. Data shown in Figure 6 only prove that PLK1 inhibition impairs tumor growth irrespective of FBXO45, IGF2BP1 or IGF2BP1 ubiquitination. How would this provide any new insights?

10. The model presented in Figure 7G provides a promising hypothesis, but the presented studies fail in conclusively supporting this ubiquitination-dependent stimulation of pro-oncogenic roles of IGF2BP1.

11. The authors should analyze whether Skp1 and Cullin1 are required for the ubiquitylation of IGF2BP1; for instance, by using a Cullin-1 dominant negative and/or a NEDD8ylation inhibitor.

12. In the experiments presented here, the authors show that an FBXO45 mutant lacking its F-box is unable to bind IGF2BP1. In contrast, most F-box protein substrates would not be affected by this deletion. Does this mean that IGF2BP1 interacts with FBXO45 at the F-box domain and not the Sprouty domain? Known substrates of SCF FBXO45 could be used as a controls for these experiments.

13. Another possibility is that FBXO45 forms a complex with the E3 ubiquitin ligase PAM, as previously shown by Saiga et al. 2020. In this case, have the authors looked whether the ubiquitylation of IGF2BP1 is dependent on PAM activity? Have the authors attempted to reconstitute this complex in vitro and undertake in vitro ubiquitylation experiments?

14. The composition and architecture of the ubiquitin chain should also be studied in more detail. The evidence provided here suggests that IGF2BP1 K190/450 sited are ubiquitylated with K63 linked chains; however, additional experiments would be required. This could include sensitivity of the ubiquitin chain to different recombinant DUBs with pan, K48, or K63 activity.

15. The authors report on a novel mouse strain that over-expresses FBXO45; however, it has not been shown whether this strain produces higher protein levels. The authors should analyze FBXO45 levels in different tissues. It is also unclear whether higher FBXO45 protein levels would translate into more active FBXO45. How is the activity of this protein regulated in cells?

16. Mass spectrometry raw data should be made available.

---

## [Author Response]

Essential revisions:The authors need to address the below raised concerns:1. The overexpression of FBXO45 in the liver needs to be demonstrated at the protein level and evaluated in respect to impact independent of DEN/CCl4 treatment.

Thanks for this Reviewer’s suggestion. We have added the protein expression level of FBXO45 in the liver tissues independent on DEN/CCl4 treatment in the revised manuscript (New data shown in Figure 2—figure supplement 1B). The result indicated that FBXO45 is highly expressed in liver in FBXO45-OE mice compared with wild type mice. In addition, FBXO45 mainly high expressed in liver compared with other tissues. Furthermore, we observed FBXO45-OE mice for up to 350 days in the absence of tumor promoters (e.g. DEN and CCl4) and found that FBXO45 significantly promoted liver fibrosis and upregulated fibrotic markers compared to wild-type mice (Author response image 1). As hepatocarcinogenesis is a complex process involving chronic liver injury, inflammation, subsequent fibrosis and carcinogenesis, it makes sense that FBXO45 may induce liver tumorigenesis in the longer-term. Here, we used DEN/CCl4 treatment just to shorten observation period. In this study, under tumor promotes DEN and CCl4 condition, FBXO45 significantly promoted liver fibrosis and hepatocarcinogenesis compared with wild-type mice.

**Author response image 1. sa2fig1:** 

2. The analysis of liver tissue (Figure 2) need further specification. Were the analyses performed on tumor tissue or non-malignant liver tissue. How was tumor content considered?

We have followed up this Reviewer’s suggestion and further specified the analysis of liver tissue in the revised manuscript. Total RNA was isolated from whole-liver samples in Figure 2D, 2F, 2G. Based on the data shown in Figure 2, FBXO45 overexpression liver tumors content contains HSC activation and hepatocyte inflammation.

3. The analyses of FBXO45 overexpression/depletion in liver cancer cell models remain puzzling. The DepMap portal clearly indicates a median positive essentiality score for FBXO45 in liver and other cancer cells (specifically in HepG2!). The authors report the opposite here based on overexpression and a single depletion without convincing control of depletion efficiency – barely any depletion is seen in Figure S2C. Here deletion studies and comparison with DepMap reported data is required.

Thank for this Reviewer’s comment on our data. The data shown in DepMap portal indicate a median positive essentiality score for *FBXO45* in liver and other cancer cells at mRNA level (Author response image 2). In addition, the CRISPR knockout of *FBXO45* has different effect on cell proliferation in various liver cancer cells (Author response image 3) , but the DepMap portal does not show the testing time points. In this study, we showed that FBXO45 at protein levels were significantly associated with liver cancer cell proliferation, as indicated by overexpression and deletion of FBXO45 for 48 hrs for CCK8 assay and 9-13 days for colony formation assay (Figure 2I-K and Figure 2—figure supplement 1E-G). The original data Figure S2C was repeated again and replaced with new data shown in Figure 2—figure supplement 1E of the revised manuscript. The data shows that FBXO45 is significantly depleted after siRNA transfection.

**Author response image 2. sa2fig2:** Expression level of FBXO45 at mRNA in various cancers and liver cancer cell lines (DepMap data).

**Author response image 3. sa2fig3:** The gene effect of *FBXO45* on cell proliferation in various cancers and liver cancer cell lines (DepMap data).

4. The data on the ubiquitination of IGf2BP1 need further in detail evaluation and consideration. In Biogrid, ubiquitination of at least 35 lysine residues, incl. the here proposed have been reported. However, no association has been reported for FBXO45, but other E3 ubiquitin ligases. Although the data appear convincing, how can the data exclude additional/other lysine residues to be modified. Moreover, the authors need to evaluate the quantity of ubiquitinated IGF2BP1 to conclusively demonstrate if this modification could contribute substantially. Along these lines, the authors need to investigate how the modification of lysine residues influences IGF2BP1-controlled mRNA turnover/translation and/or RNA-association.

It is true that IGF2BP1 has ubiquitination at least 35 lysine residues. Here, we identified the whole ubiquitinated proteins by ubiquitome using FBXO45 tumors and normal tissues(data shown in Figure 3A). We found that IGF2BP1 only exhibited the upregulated ubiquitination at Lys190 and Lys450 sites in FBXO45 overexpression tumors compared with normal tissues. Next, we wanted to know whether these two sites were involved in FBXO45-mediated IGF2BP1 polyubiquitination. Our results indicated FBXO45 enhanced IGF2BP1 polyubiquitination, while the K190A or K450A mutants significantly blocked this action (Figure 3E), suggesting that K190 and K450 are responsible for FBXO45-mediated IGF2BP1 polyubiquitination. Furthermore, we found that combination of the ubiquitination level of K190A and K450A groups is similar to total ubiquitination level of IGF2BP1, which further supported that FBXO45 promoted IGF2BP1 ubiquitination at K190 and K450 sites.

RIP-PCR analysis indicated that IGF2BP1 interacted with *PLK1* mRNA in both HepG2 and HCCLM3 cells (New data shown in Figure 4E and Figure 4—figure supplement 1A). Inhibition of RNA polymerase by actinomycin D (act D) significantly reduced *PLK1* and *PEG10* mRNA accumulation (*PEG10* is a well-known target mRNA stabilized by IGF2BP1) and silencing of IGF2BP1 further enhanced this action (New data shown in Figure 4F and Figure 4—figure supplement 1B). Taken together, these results demonstrated that IGF2BP1 bound *PLK1* mRNA and promoted *PLK1* mRNA stability. As the ubiquitination of IGF2BP1 at K190 and K450 sites related to its activity(Figure 3H), we can draw a conclusion that FBXO45 promoted IGF2BP1 polyubiquitination at K190 and K450 sites, which enhanced its binding with *PLK1* mRNA and promoted *PLK1* mRNA stability.

5. Although PLK1 likely is a target mRNA stabilized by IGF2BP1, the data presented in Figure 4C-E are only partially convincing. The steady state mRNA levels of PLK1 are only reduced by ~20%, whereas the protein content suggests a substantially more pronounced downregulation. Moreover, in one cell line FOXK1 is down – whereas it is upregulated in the other cell line upon IGF2BP1 depletion. Finally, the decay analyses (Figure 4E) require more time points, semi-logarithmic data presentation and the error bars are unlikely to result from biological but rather technical replicates. Additional positive and negative controls are required here. Finally, establishing IGF2BP1-dependent regulation of PLK1 expression requires investigation of IGF2BP1-PLK1 mRNA association and need to include studies on how IGF2BP1 depletion impairs PLK1-directed phosphorylation.

Thanks for this Reviewer’s suggestion. In order to enhance the knockdown efficiency of IGF2BP1, we have used lentivirus-mediated shRNA to knockdown IGF2BP1 expression in both HCCLM3 and HepG2 cells. The new data shown in Figure 4C-D have demonstrated that IGF2BP1 silencing significantly inhibited the expression of IGF2BP1, PLK1 and, KI67 at mRNA and protein levels in both HepG2 and HCCLM3 cells. The results are consistent in these two cells.

As for the decay analyses (Figure 4E in the original version), we have followed up this Reviewer’s suggestion and added more time points, repeated more than 3 times and added a positive control (*PEG10*, a well-known target mRNA stabilized by IGF2BP1) and a negative control (*β-actin*) (Zhang et al. Theranostics, 2021). As shown in Figure 4F and Figure 4—figure supplement 1B of the revised manuscript, IGF2BP1 promoted *PLK1* and *PEG10* mRNA stability but not *β-actin*.

Regarding IGF2BP1-PLK1 mRNA association, we have done the RIP-PCR analysis and the new data indicated that IGF2BP1 bound to *PLK1* mRNA in both HepG2 and HCCLM3 cells (New data shown in Figure 4E and Figure 4—figure supplement 1A). Inhibition of RNA polymerase by actinomycin D (act D) significantly reduced *PLK1* and *PEG10* mRNA accumulation (*PEG10* is a well-known target mRNA stabilized by IGF2BP1) and silencing of IGF2BP1 further enhanced this action (New data shown in Figure 4F and Figure 4—figure supplement 1B). Taken together, these results demonstrated that IGF2BP1 bound *PLK1* mRNA and promoted *PLK1* mRNA stability to upregulation of PLK1 expression. Here we did not talk about PLK1 phosphorylation in the original version.

6. Figure 4F does not show a dramatic attenuation of FBXO45-promoted cell proliferation. Instead, it only demonstrates that IGF2BP1 depletion results in modestly reduced proliferation in both, wild type as well as FBXO45-overexpressing cells. How does depletion affect FBXO45 expression?

Maybe the siRNA targeting IGF2BP1 used in the previous version was not strong. Here we have used shIGF2BP1 lentivirus against IGF2BP1 expression in primary hepatocytes. As new data shown in Figure 4D of the revised manuscript, knockdown of IGF2BP1 by lentivirus could significantly reduce IGF2BP1 expression. We found that knockdown of IGF2BP1 by shIGF2BP1 dramatically reduced cell proliferation in both wild type and FBXO45-overexpressing cells (New data shown in Figure 4G of the revised manuscript). The data presented in Figure 4G are convincing.

7. Analyses of PLK1 inhibition do not provide any useful insights. Here it needs to be demonstrated that IGF2BP1 deletion and FBXO45 overexpression modulate the potency of the respective inhibitors and that this is connected to IGF2BP1 ubiquitination.

We found that FBXO45 upregulated PLK1 accumulation in HCCLM3 cells and knockdown of IGF2BP1 attenuated this action, suggesting FBXO45 increased PLK1 expression via IGF2BP1 (Author response image 4). As IGF2BP1 activated by FBXO45-mediated polyubiquitination(Figure 3H), it makes sense that IGF2BP1 ubiquitination requires for *PLK1* mRNA stability and PLK1 protein level accumulation. This was supported by FBXO45 significantly promoted wild type IGF2BP1-mediated cell proliferation but not its mutants (K190A and K450A) (Figure 3I).

**Author response image 4. sa2fig4:** 

8. Results presented in Figure 5 provide interesting hints for a role of IGF2BP1 in promoting HCC, as previously reported, and regulation of PLK1 expression, but they fail to demonstrate that this is linked to FBXO45. Why is FBXO45 expression not monitored here by Western blotting?

We have followed up this reviewer’s constructive suggestion and added the expression of FBXO45 in Figure 5D of the revised manuscript.

9. Data shown in Figure 6 only prove that PLK1 inhibition impairs tumor growth irrespective of FBXO45, IGF2BP1 or IGF2BP1 ubiquitination. How would this provide any new insights?

As shown in Figure 6, FBXO45 overexpression enhanced tumor growth and this effect could be significantly attenuated by PLK1 inhibitor, suggesting inhibition of PLK1 could be a potential target for HCC patients with FBXO45 high expression. Our previous data have demonstrated that FBXO45 upregulated PLK1 expression via IGF2BP1 ubiquitination and activation (Data shown in Figure 3E-I and Figure 4C-D).

10. The model presented in Figure 7G provides a promising hypothesis, but the presented studies fail in conclusively supporting this ubiquitination-dependent stimulation of pro-oncogenic roles of IGF2BP1.

Our pervious data showed that FBXO45 promoted wild-type IGF2BP1 ubiquitination and activation(Figure 3E-F and Figure 3—figure supplement 1D), as indicated by low pro-oncogenic role of IGF2BP1 with K190 and K450 mutants on cell proliferation compared with wild-type IGF2BP1 (Figure 3H). In addition, K63-linked ubiquitination is usually related to protein activation or stability. Here, we found that FBXO45 mediated K63-linked polyubiquitination of IGF2BP1 and then activation(Figure 3F). Our data support ubiquitination-dependent stimulation of pro-oncogenic roles of IGF2BP1.

11. The authors should analyze whether Skp1 and Cullin1 are required for the ubiquitylation of IGF2BP1; for instance, by using a Cullin-1 dominant negative and/or a NEDD8ylation inhibitor.

Many E3 ubiquitin ligases contain a Cullin scaffolding protein, a Skp adaptor protein, and an F-box protein. However, FBXO45 acts as an atypical E3 ubiquitin ligase, which contains Skp1 but lacks a Cullin (Saiga et al. Mol Cell Biol 2009; Brace et al. J neurosic 2014). Consistently, according to the immunoprecipitation (IP)-MS results, FBXO45, upon ectopic expression in HCCLM3 cells, pulled down endogenous SKP1 but without Cullin-1. In line with this finding, our new data shown in Figure 3—figure supplement 2B demonstrates that FBXO45 binds to SKP1 but not endogenous Cullin-1. These results indicated FBXO45-mediated IGF2BP1 ubiquitylation requires Skp1 but not requires Cullin-1.

12. In the experiments presented here, the authors show that an FBXO45 mutant lacking its F-box is unable to bind IGF2BP1. In contrast, most F-box protein substrates would not be affected by this deletion. Does this mean that IGF2BP1 interacts with FBXO45 at the F-box domain and not the Sprouty domain? Known substrates of SCF FBXO45 could be used as a controls for these experiments.

Here we have chosen N-cadherin (a FBXO45 target protein binding at the Sprouty domain) as a positive control(Chung et al. J Biol Chem 2014). The new data shown in Figure 3—figure supplement 1A demonstrated that FBXO45 and FBXO45ΔF (FBXO45 without F-box domain) could pulldown N-cadherien, suggesting N-cadherin didn’t bound FBXO45 at F-box domain. However, FBXO45 but not FBXO45ΔF could pulldown IGF2BP1(Figure 3D), indicating that IGF2BP1 interacts with FBXO45 at the F-box domain.

13. Another possibility is that FBXO45 forms a complex with the E3 ubiquitin ligase PAM, as previously shown by Saiga et al. 2020. In this case, have the authors looked whether the ubiquitylation of IGF2BP1 is dependent on PAM activity? Have the authors attempted to reconstitute this complex in vitro and undertake in vitro ubiquitylation experiments?

We have followed up this Reviewer’s constructive suggestion and analyzed whether PAM was involved in FBXO45-mediated IGF2BP1 ubiquitination. As new data shown in Figure 3—figure supplement 2A of the revised manuscript, knockdown of PAM could significantly attenuated FBXO45-mediated IGF2BP1 ubiquitination, suggesting he ubiquitylation of IGF2BP1 is dependent on PAM activity.

14. The composition and architecture of the ubiquitin chain should also be studied in more detail. The evidence provided here suggests that IGF2BP1 K190/450 sited are ubiquitylated with K63 linked chains; however, additional experiments would be required. This could include sensitivity of the ubiquitin chain to different recombinant DUBs with pan, K48, or K63 activity.

Our previous result indicated that K63 mutant (K63R) could significantly blocked IGF2BP1 polyubiquitination compared with wild-type and K48 mutant ubiquitin group. In order to further validate this finding, we investigated the role of wild-type Ub, K63, and K48 Ub in FBXO45-mediated IGF2BP1 ubiquitination. We found that K63 Ub- mediated IGF2BP1 ubiquitination was similar to wild-type Ub group, and they were all stronger than K48 Ub group (New data shown in Figure 3—figure supplement 1D). Taken together, these results indicated that FBXO45 promoted IGF2BP1 via K63-linked ubiquitination.

15. The authors report on a novel mouse strain that over-expresses FBXO45; however, it has not been shown whether this strain produces higher protein levels. The authors should analyze FBXO45 levels in different tissues. It is also unclear whether higher FBXO45 protein levels would translate into more active FBXO45. How is the activity of this protein regulated in cells?

Thanks for this Reviewer’s suggestion. We have detected the FBXO45 expression by IHC staining and WB. The new data shown in Figure 5E, 6D and Figure 2—figure supplement 1B indicated the higher expression of FBXO45 in overexpressed FBXO45 liver tissues. Also, we have analyzed FBXO45 levels in different tissues and found that FBXO45 is highly expressed in liver tissues compared with other tissues (i.g. brain, spleen, kidney, lung and skin) (New data shown in Figure 2—figure supplement 1C of the revised manuscript). Based on our in vivo and in vitro cell proliferation assay, it indicated that high expression of FBXO45 could translate into more active FBXO45 and promote tumor growth and cell proliferation (Figure 2A , 2I-K, Figure 2—figure supplement 1E-G).

16. Mass spectrometry raw data should be made available.

We have followed up this suggestion and added mass spectrometry raw data in the revised manuscript, named as "Figure 3-source data 1" and Supplementary file 4.

References:

Zhang L, et al., IGF2BP1 overexpression stabilizes PEG10 mRNA in an m6A-dependent manner and promotes endometrial cancer progression. Theranostics 11 1100-1114 (2021).

Saiga, T. et al., Fbxo45 forms a novel ubiquitin ligase complex and is required for neuronal development. Mol Cell Biol 29 3529 (2009).

Brace, E. J., Wu, C., Valakh, V. and DiAntonio, A., SkpA restrains synaptic terminal growth during development and promotes axonal degeneration following injury. J Neurosci 34 8398 (2014).

Chung, F. Z. et al., Fbxo45 inhibits calcium-sensitive proteolysis of N-cadherin and promotes neuronal differentiation. J Biol Chem 289 28448 (2014)